# DreamClean: Restoring Clean Image Using Deep Diffusion Prior

**Jie Xiao**[1]  **Ruili Feng**[2]  **Han Zhang**[3]  **Zhiheng Liu**[1]  **Zhantao Yang**[3]

**Yurui Zhu**[1]  **Xueyang Fu**[1†]  **Kai Zhu**[2]  **Yu Liu**[2]  **Zheng-Jun Zha**[1]

[1]University of Science and Technology of China   [2]Alibaba Group
[3]Shanghai Jiao Tong University
ustchbxj@mail.ustc.edu.cn ruilifengustc@gmail.com xyfu@ustc.edu.cn

## Abstract

Image restoration poses a garners substantial interest due to the exponential surge in demands for recovering high-quality images from diverse mobile camera devices, adverse lighting conditions, suboptimal shooting environments, and frequent image compression for efficient transmission purposes. Yet this problem gathers significant challenges as people are blind to the type of restoration the images suffer, which, is usually the case in real-day scenarios and is most urgent to solve for this field. Current research, however, heavily relies on prior knowledge of the restoration type, either explicitly through rules or implicitly through the availability of degraded-clean image pairs to define the restoration process, and consumes considerable effort to collect image pairs of vast degradation types. This paper introduces DreamClean, a training-free method that needs no degradation prior knowledge but yields high-fidelity and generality towards various types of image degradation. DreamClean embeds the degraded image back to the latent of pre-trained diffusion models and re-sample it through a carefully designed diffusion process that mimics those generating clean images. Thanks to the rich image prior in diffusion models and our novel Variance Preservation Sampling (VPS) technique, DreamClean manages to handle various different degradation types at one time and reaches far more satisfied final quality than previous competitors. DreamClean relies on elegant theoretical supports to assure its convergence to clean image when VPS has appropriate parameters, and also enjoys superior experimental performance over various challenging tasks that could be overwhelming for previous methods when degradation prior is unavailable.

## 1 Introduction

Image Restoration (IR), which is a classic ill-posed inverse problem, aims to recover a clean version from a degraded observation. Currently, deep learning-based IR techniques have demonstrated promising performance and dominated this field, which could be broadly categorized into supervised and unsupervised paradigms.

Supervised-based IR solutions usually rely on large-scale pre-collected paired datasets to train their models. A major challenge is that they implicitly assume training and testing data should be identically distributed. As a result, these methods often deteriorate seriously in performance when testing cases deviate the pre-assumed distribution. In addition, once the underlying degradation model is changed, a new dataset needs be re-collected and a new model has to be re-trained, which can be both time-consuming and costly.

Another prevailing research line is unsupervised-based IR approaches. They explicitly make use of the degradation model to produce a clean image by solving a maximum a posterior problem or

---

† Corresponding author

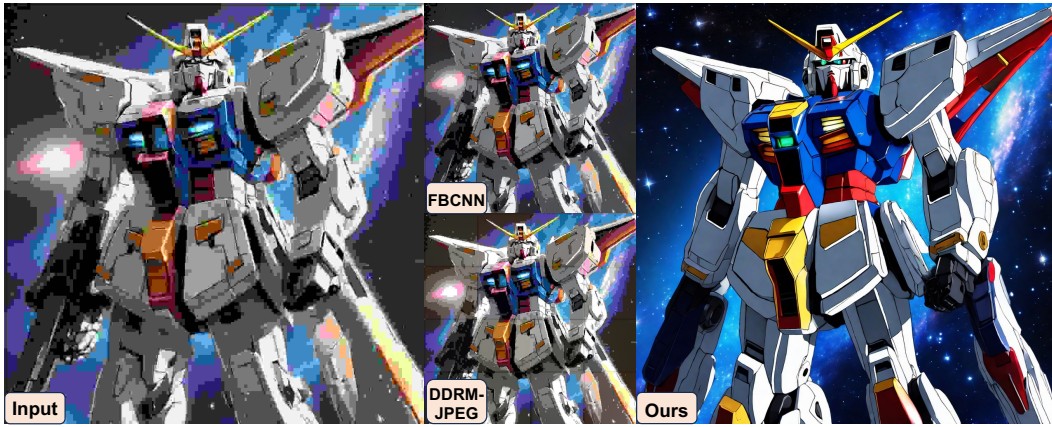

Figure 1: Results of JPEG artifacts correction. The image is degraded by multiple non-align JPEG compression with $\text{QF} = \{5, 10, 20\}$ and shift $\{0, 3, 6\}$. FBCNN is a supervised method and DDRM-JPEG is an unsupervised solution using the worst $\text{QF} = 5$ as the degradation model. Our Dream-Clean is blind to the degradation model. DreamClean can still recover a $1024 \times 1024$ high-quality image given the extremely destroyed image based on the advanced Stable Diffusion XL.

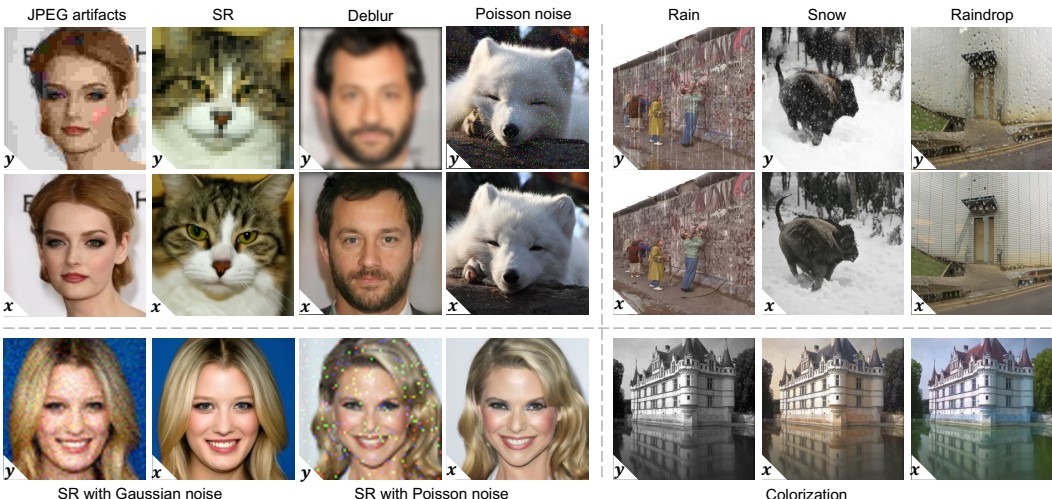

Figure 2: We propose DreamClean to solve various image restoration problems *without* task-specific re-training *or* assuming the known degradation model. DreamClean can resort to the inherent prior of diffusion models to tackle with linear degradation, noisy linear degradation, non-linear degradation and complex bad weather degradation. $\boldsymbol{y}$: the degraded image, $\boldsymbol{x}$: our result.

a posterior sampling problem. For example, DDRM (Kawar et al., 2022a) hypothesizes the linear degradation model and relies on the desirable property of linear formulation to sample from posterior distribution. In practice, however, the underlying degradation model may be too complex to estimate or computationally prohibitive to apply (Ongie et al., 2020). In addition, these approaches may not be ready to be equipped with diffusions trained in VAE-encoded space (Rombach et al., 2022) since VAE projection may complicate the entanglement between degraded and clean information [1].

To release the generative power of diffusion models from the heavy degradation prior, we propose a novel training-free and unsupervised framework, dubbed DreamClean, for general IR problems. DreamClean bypasses the requirement of paired dataset and can generate samples without explicit or implicit assumptions about the specific degradation model, resulting in strong robustness to vast degradation types. As shown in Figure 2, DreamClean can tackle with various types of degradation, ranging from typical linear degradation (image coloration, super-resolution, deblurring), noisy linear

---

[1] *e.g.*, the linear form $\boldsymbol{y} = \boldsymbol{Hx}$ of the degradation model in pixel space will not hold in the encoded space.

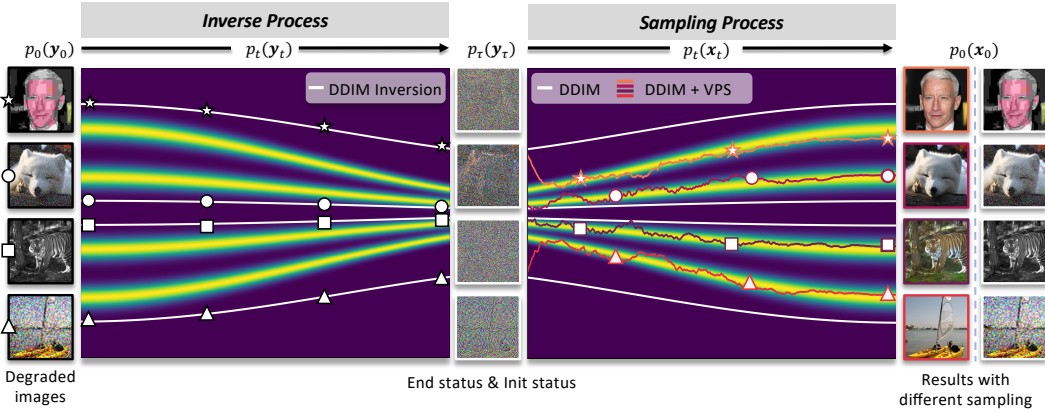

Figure 3: Overview of DreamClean. DDIM and its inversion can reconstruct the input image, thus providing informative latents. VPS can guide low-probability latents to move towards vicinal high probability region, which produces clean image samples while maintaining similarity with input degraded images. (Best viewed on screen.)

degradation (Poisson noise, SR with Gaussian and Poisson noise), non-linear degradation (multiple non-align JPEG artifacts correction (Jiang et al., 2021)), to complex bad weather degradation (rain, snow, raindrop). DreamClean works by, like an experienced human, "imagining" the potential clean image purely based on an input degraded observation.

The key idea behind DreamClean is to search in clean image distribution, which is represented by a diffusion prior, to find the clean image while being faithful to the input degraded image. Consequently, the first core ingredient of our framework is a pre-trained diffusion model. We treat such a diffusion model as a solution for an extreme IR problem: it can generate clean images *even if* all information about the clean image is lost [2]. Another key issue to be addressed is to ensure faithfulness to the degraded image. We resort to the inversion of ODE sampling algorithm (*e.g.*, DDIM (Song et al., 2021a)) of diffusion model to accomplish this goal. As illustrated in Figure 3, through reconstructing the degraded image, DDIM inversion algorithm can produce a series of latents which preserve information about the input image. These latent variables locate in low-probability region since sampling from the diffusion model generally produces clean images rather than degraded ones. Although these latents cannot restore the clean image directly, they can inherit information from the input image, providing good initializations for subsequent sampling. Inspired by this, we propose Variance Preservation Sampling (VPS) to guide these corrupted low-probability latents towards nearby high-probability region from which clean samples can be generated. In this way, VPS functions as a general solution to ensure faithfulness even without knowing the specific degradation model. It is also noteworthy that i) DreamClean does not assume specific form for the underlying degradation model. Therefore, it can be integrated with diffusion models pre-trained in pixel space as well as VAE-encoded space. As shown in Figure 1, DreamClean can still accomplish the challenging multiple non-align JPEG artifacts correction when applied in the encoded space of Stable Diffusion XL (Podell et al., 2023); ii) DreamClean is orthogonal to previous works which exploit the degradation model to sample from posterior distribution. DreamClean can also make use of the degradation model to produce more faithful results. Our method enjoys both elegant theoretical guarantees in convergence and superior performance in many challenging scenarios.

## 2 METHOD

### 2.1 PRELIMINARY

Denoising diffusion probabilistic models (DDPMs) are latent variable models aiming to learn a model distribution $p(x_0)$ to approximate the data distribution $q(x_0)$ (Ho et al., 2020). DDPMs comprise $T$-step forward diffusion process, which disturbs data by slowly adding Gaussian noise and $T$-step reverse generative process, which samples data by progressively removing noise. The

---

[2] A diffusion model can generate a clean image from a standard Gaussian noise.

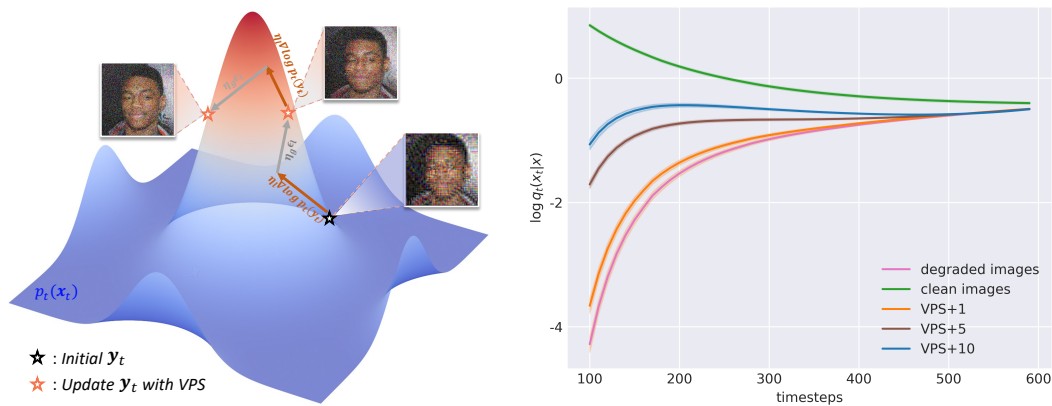

Figure 4: Illustration of the proposed Variance Preservation Sampling algorithm.

Figure 5: VPS drives latent variables to high probability region.

forward process is a Markov chain which is of the form

$$q(\boldsymbol{x}_{1:T}|\boldsymbol{x}_0) := \prod_{t=1}^{T} q(\boldsymbol{x}_t|\boldsymbol{x}_{t-1}), \quad q(\boldsymbol{x}_t|\boldsymbol{x}_{t-1}) := \mathcal{N}(\boldsymbol{x}_t; \sqrt{1-\beta_t}x_{t-1}, \beta_t \boldsymbol{I}) \tag{1}$$

where $\{\beta_t\}_{t=0}^{T}$ is the variance schedule. $\{\boldsymbol{x}_t\}_{t=0}^{T}$ are latent variables, which we refer to as latents below. A property of diffusion process is that the conditional distribution of $x_t$ given $x_0$ is of form

$$q(\boldsymbol{x}_t|\boldsymbol{x}_0) = \mathcal{N}(\boldsymbol{x}_t; \sqrt{\bar{\alpha}_t}\boldsymbol{x}_0, (1-\bar{\alpha}_t)\boldsymbol{I}), \quad \text{where} \quad \alpha_t = 1 - \beta_t, \bar{\alpha}_t = \prod_{i=0}^{t} \alpha_t. \tag{2}$$

The reverse generative process proceeds by sampling from a Markov chain starting at Gaussian noise $p(\boldsymbol{x}_T) = \mathcal{N}(\boldsymbol{x}_T; \boldsymbol{0}, \boldsymbol{I})$

$$p_\theta(\boldsymbol{x}_{t-1}|\boldsymbol{x}_t) \approx q(\boldsymbol{x}_{t-1}|\boldsymbol{x}_t, \boldsymbol{x}_0) = \mathcal{N}(\boldsymbol{x}_{t-1}; \boldsymbol{\mu}_\theta(\boldsymbol{x}_t, \boldsymbol{x}_0), \sigma_t^2 \boldsymbol{I}), \tag{3}$$

with reparameterization, $\boldsymbol{\mu}_\theta(\boldsymbol{x}_t, \boldsymbol{x}_0)$ and $\sigma_t^2$ have the closed form

$$\boldsymbol{\mu}_\theta(\boldsymbol{x}_t, \boldsymbol{x}_0) = \frac{1}{\sqrt{\alpha_t}} \left( \boldsymbol{x}_t - \frac{\beta_t}{\sqrt{1-\bar{\alpha}_t}} \boldsymbol{\epsilon}_\theta(\boldsymbol{x}_t, t) \right), \quad \sigma_t^2 = \frac{1-\bar{\alpha}_{t-1}}{1-\bar{\alpha}_t} \beta_t. \tag{4}$$

Song et al. (2021b) demonstrate the aforementioned reverse process is a discretization of a continuous-time stochastic process, described by the following reverse-time stochastic differential equation (SDE)

$$d\boldsymbol{x}_t = \left[ f(t)\boldsymbol{x}_t - g^2(t)\nabla_{\boldsymbol{x}_t} \log p_t(\boldsymbol{x}_t) \right] dt + g(t)d\bar{\boldsymbol{w}}_t, \tag{5}$$

where $\bar{\boldsymbol{w}}_t$ is a standard Wiener process in the reverse time, $f(t) = \frac{1}{2}\frac{d \log \bar{\alpha}(t)}{dt}$, $g(t) = (1 - \bar{\alpha}(t))\frac{d}{dt}\frac{1-\bar{\alpha}(t)}{\bar{\alpha}(t)}$, and $\bar{\alpha}(t)$ is a continuous version of $\bar{\alpha}_t$. For the reverse-time SDE, Song et al. (2021b) further prove that there exists a corresponding probability flow ODE that shares the same marginal distribution:

$$d\boldsymbol{x}_t = \left[ f(t)\boldsymbol{x}_t - \frac{1}{2}g^2(t)\nabla_{\boldsymbol{x}_t} \log p_t(\boldsymbol{x}_t) \right] dt. \tag{6}$$

With this probability flow ODE, one can generate an image from a Gaussian noise and vice versa.

## 2.2 DreamClean

DreamClean focuses on exploiting image priors captured by diffusion models pre-trained on large-scale diverse-distributed images. DreamClean restores a degraded image $\boldsymbol{y}$ by finding a clean sample $\boldsymbol{x}$ which simultaneously satisfies i) it is faithful to the degraded image; ii) it conforms model distribution of pre-trained diffusion models. Below are our strategies towards these constraints.

**Faithfulness by ODE Inversion.** ODE sampling algorithm is approximately invertible, that is, for a given image, one can find a series of latents, any of which can reproduce the input image along

the ODE sampling trajectory (as shown in Figure 3). This property implies that these inverse latents should contain desirable information about the input image. On the other hand, high quality images are generated when we sample from pre-trained diffusion models, which means these latents lie in low probability region. Inspired by this, as illustrated in Figure 3, we propose to utilize the inverse latents as initialization and design a correcting algorithm to guide these latents towards nearby high probability region. In this work, we choose DDIM (Song et al., 2021a) as the default ODE sampling. Assume the degraded image is $\boldsymbol{y}$, we can find a latent $\boldsymbol{y}_\tau$ by the DDIM inversion

$$\boldsymbol{y}_\tau = \mathrm{DDIM}^{-1}\left(\boldsymbol{y}\right), \tag{7}$$

where $\mathrm{DDIM}^{-1}\left(\cdot\right)$ is the inversion of DDIM and $0 < \tau \leq T$ denotes the strength of ODE inverse.

**Realness by Variance Preservation Sampling.** After getting the informative latent $\boldsymbol{y}_\tau$, We can correct the low-probability latent and gradually denoise it to get the clean image. We conduct the following two steps at each timestep $t \in [\tau, 0)$

$$\begin{aligned} \boldsymbol{y}_t^m &= \boldsymbol{y}_t^{m-1} + \eta_l \nabla \log p_t(\boldsymbol{y}_t^{m-1}) + \eta_g \boldsymbol{\epsilon}_g^m, &\text{(Variance Preservation Sampling)} \\ m &= 1, \cdots, M, \boldsymbol{y}_t^0 = \boldsymbol{y}_t, \\ \boldsymbol{y}_{t-1} &= \mathrm{DDIMStep}(\boldsymbol{y}_t^M), &\text{(Denoise Sampling)} \end{aligned} \tag{8}$$

where $\eta_l$ and $\eta_g$ are required to satisfy the constraint:

$$\eta_l = \gamma(1 - \bar{\alpha}_t), \ \eta_g = \sqrt{\gamma(2 - \gamma)}\sqrt{1 - \bar{\alpha}_t}. \tag{9}$$

$0 < \gamma < 1$ is a scalar determining the step size and $\bar{\alpha}_t$ is the noise schedule defined in Equation (2). Such setting of $\eta_l$ and $\eta_g$ is vital to restore a clean image, which we will discuss later in Theorem 2.2. Intuitively, as shown in Figure 4, given the initial latent which lies in low-probability region, VPS guides the latent to move towards its vicinal high-probability region. The high-probability region conforms the normal sampling formulation of diffusion models. Therefore, by correcting latents progressively, VPS can produce high quality images. In practice, $\nabla \log p(\boldsymbol{y}_t^{m-1})$ can be computed by a pre-trained diffusion models (Hyvärinen & Dayan, 2005; Karras et al., 2022). Specifically, the gradient term has the relation with the predicted noise by a pre-trained diffusion model $\boldsymbol{\epsilon}_\theta$:

$$\nabla \log p_t\left(\boldsymbol{y}_t^{m-1}\right) = -\frac{\boldsymbol{\epsilon}_\theta\left(\boldsymbol{y}_t^{m-1}, t\right)}{\sqrt{1 - \bar{\alpha}_t}}. \tag{10}$$

We argue that when $\eta_l$ and $\eta_g$ is subject to Equation (9), VPS converges to a nearby high probability set, which in turn generates a potential clean image $\boldsymbol{x}$. Since latents of diffusion models are typical of high dimensionality, inspired by the concept of typical set in information theory (Shannon, 1948; Cover & Thomas, 2006), we define the set that gathers most density of $\boldsymbol{x}_0$-induced latents as the following High Probability Set, where $\boldsymbol{x}_0$ is a clean image.

**Definition 2.1** (High Probability Set). *For $\delta > 0$, $t > 0$ and potential clean image $\boldsymbol{x}_0 \in \mathbb{R}^N$, High Probability Set $\mathcal{T}_t^N\left(\boldsymbol{x}_0; \delta\right)$ is defined as follows*

$$\mathcal{T}_t^N\left(\boldsymbol{x}_0; \delta\right) = \left\{\boldsymbol{x}_t : \left|-\frac{1}{N}\log p_t\left(\boldsymbol{x}_t | \boldsymbol{x}_0\right) - H\right| \leq \delta\right\}. \tag{11}$$

*where $p_t\left(\boldsymbol{x}_t | \boldsymbol{x}_0\right) = \prod_{i=1}^N p_t\left(x_{t,i} | x_{0,i}\right)$, $x_{t,i}$ and $x_{0,i}$ denote the $i$-th elements of $\boldsymbol{x}_t$ and $\boldsymbol{x}_0$ respectively, and $H$ is the Shannon entropy of $\mathcal{N}(0, 1 - \bar{\alpha}_t)$.*

According to law of large numbers, the probability of $\boldsymbol{x}_t \in \mathcal{T}_t^N\left(\boldsymbol{x}_0; \delta\right)$ gets close to 1 for sufficiently large $N$ (see Appendix A.2 for details). In other words, $\mathcal{T}_t^N\left(\boldsymbol{x}_0; \delta\right)$ is a set comprising of latents of a clean image $\boldsymbol{x}_0$ whose probability can be sufficiently large. We can prove that VPS drives latents of degraded images to High Probability Set of a nearby clean image under appropriate $\eta_l, \eta_g$.

**Theorem 2.2.** *For $\delta > 0$ and the potential image $\boldsymbol{x}_0$, when $\eta_l$ and $\eta_g$ satisfy the constraint in Equation (9), there exists an inverse time $\tau$ such that $\boldsymbol{y}_\tau^M$ by Variance Preservation Sampling converges to $\mathcal{T}_\tau^N\left(\boldsymbol{x}_0; \delta\right)$ when $M \to \infty$.*

We delay proof to Appendix A.1. Satisfying Equation (9) is vital for clean image restoration, which is demonstrated theoretically in Appendix A.1 and empirically in Section 3.6. Theorem 2.2 implies that at timestep $\tau$, VPS is capable of correcting latents of a degraded image to a nearby High Probability Set, which then generates the clean image following sampling dynamics of diffusion models.

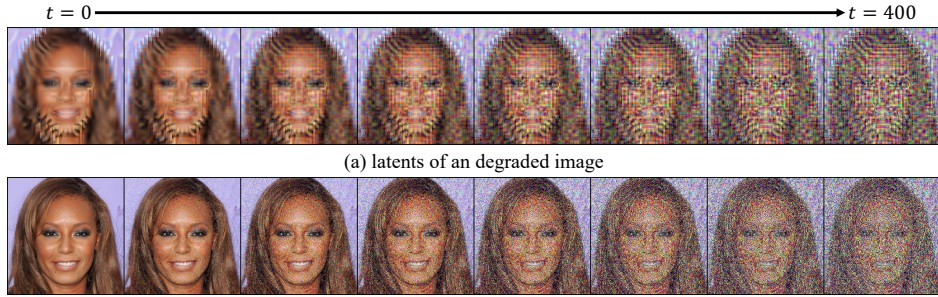

$t = 0 \longrightarrow t = 400$

(a) latents of an degraded image

(b) VPS boosted latents

Figure 6: Visualization of latents of different timesteps. VPS changes original degraded artifacts to Gaussian-like noise, which conforms the formulation of diffusion models.

In implementation, we fix the inverse strength $\tau$ (*e.g.*, 300 for $T = 1000$), and take 1-step VPS correction before each DDIM step.

**Put them together.** In conclusion, DDIM inversion finds an informative latent as the initialization and VPS corrects it towards the High Probability Set of a nearby clean image. These two mechanisms cooperate to achieve the goal of realness and faithfulness.

## 3 EXPERIMENTS

### 3.1 EXPERIMENTAL SETUP

Our experiments consist of: i) verifying that DreamClean optimizes latents to higher probability region (Section 3.2); ii) quantitative comparison with previous methods (Sections 3.3 and 3.4); iii) presentation of visual results across multiple degradation types to demonstrate its strong robustness and generality (Section 3.4); iv) exploiting the degradation model (Section 3.5); We demonstrate that DreamClean is orthogonal to prior works, which can exploit the underlying degradation model to achieve challenging inverse problem (*e.g.*, phase retrieval); v) ablation study on the different schedules of $\eta_l$ and $\eta_g$.

### 3.2 MOVING TO HIGHER PROBABILITY

For the latent variable $x_t$, we evaluate its probability under a pre-trained diffusion model by $\log p_\theta (x_t)$. Given $x_0$, we can approximate it by the alternative score $\log q_t (x_t|x_0)$. We use the noisy SR as the default IR task and record the average score and the standard deviation on CelebA 1K. Figure 5 shows that latents of degraded images locate in low-probability region when compared with clean images and VPS gradually promotes their probability. We also provide a more intuitive visualization of latents with different timesteps in Figure 6. We can find that driven by VPS, latents with unexpected artifacts are transformed to the appearance of a clean image with Gaussian-like noise, which conforms the sampling dynamics (Equation (2)) of diffusion models.

### 3.3 QUANTITATIVE EXPERIMENTS

We validate the efficacy of DreamClean using the diffusion models (Ho et al., 2020; Dhariwal & Nichol, 2021) trained on CelebA (Karras et al., 2018), LSUN bedroom (Yu et al., 2015) and ImageNet (Deng et al., 2009). For quantitative comparison with previous methods, we perform experiments on the classic IR tasks, including linear degradation (noisy image super-resolution) and complex non-linear degradation (multiple non-align JPEG compression artifacts correction). We use the average peak signal-to-noise ratio (PSNR) and structural similarity index measure (SSIM) to measure faithfulness and Learned Perceptual Image Patch Similarity (LPIPS) as the perceptual metrics. Following (Kawar et al., 2022a), we also report number of function evaluations (NFEs) for each experiment to compare efficiency.

We use ImageNet 1K (Deng et al., 2009), CelebA 1K (Karras et al., 2018), and validation set of LSUN bedroom (Yu et al., 2015) with image size $256 \times 256$ for validation. We perform comparison with RED (Romano et al., 2017), DGP (Pan et al., 2021), SNIPS (Kawar et al., 2021),

Table 1: Quantitative results of $4\times$SR with Gaussian noise $\sigma = 0.05$ on CelebA.

| Method | PSNR ↑ | SSIM ↑ | LPIPS ↓ | NFEs ↓ |
|--------|--------|--------|---------|--------|
| Baseline | 23.64 | 0.51 | 0.64 | 0 |
| DGP | 18.40 | 0.40 | 0.70 | 1500 |
| SNIPS | 26.38 | 0.74 | 0.20 | 1000 |
| DPS | 24.42 | 0.70 | 0.17 | 1000 |
| DDRM | 29.21 | 0.83 | 0.09 | 100 |
| DDNM | 29.17 | 0.82 | 0.09 | 100 |
| GDP | 24.38 | 0.71 | 0.15 | 1000 |
| Ours | 27.23 | 0.77 | 0.12 | 90 |
| Ours* | **30.19** | **0.84** | **0.08** | 60 |

Table 2: Quantitative results of $4\times$SR with Gaussian noise $\sigma = 0.05$ on ImageNet.

| Method | PSNR ↑ | SSIM ↑ | LPIPS ↓ | NFEs ↓ |
|--------|--------|--------|---------|--------|
| Baseline | 21.85 | 0.47 | 0.58 | 0 |
| DGP | 9.50 | 0.12 | 0.93 | 1500 |
| RED | 22.90 | 0.49 | NA | 100 |
| DPS | 24.42 | 0.70 | 0.36 | 1000 |
| DDRM | 25.67 | 0.73 | 0.30 | 100 |
| DDNM | 25.56 | 0.72 | 0.30 | 100 |
| GDP | 24.33 | 0.67 | 0.39 | 1000 |
| Ours | 24.31 | 0.67 | 0.40 | 90 |
| Ours* | **25.84** | **0.74** | **0.23** | 60 |

Table 3: Quantitative results of JPEG compression artifacts correction on CelebA.

| Method | PSNR ↑ | SSIM ↑ | LPIPS ↓ | NFEs ↓ |
|--------|--------|--------|---------|--------|
| Baseline | 24.79 | 0.69 | 0.41 | 0 |
| QGAC | 24.28 | 0.68 | 0.32 | 1 |
| FBCNN | 26.37 | 0.77 | 0.24 | 1 |
| DDNM | 24.40 | 0.66 | 0.31 | 100 |
| DDRM-JPEG | 26.41 | 0.77 | **0.20** | 100 |
| Ours | **27.58** | **0.82** | **0.20** | 90 |

Table 4: Quantitative results of JPEG compression artifacts correction on LSUN bedroom.

| Method | PSNR ↑ | SSIM ↑ | LPIPS ↓ | NFEs ↓ |
|--------|--------|--------|---------|--------|
| Baseline | 23.39 | 0.68 | 0.34 | 0 |
| QGAC | 23.41 | 0.69 | 0.34 | 1 |
| FBCNN | 24.10 | 0.73 | **0.31** | 1 |
| DDNM | 22.73 | 0.66 | 0.33 | 100 |
| DDRM-JPEG | 24.06 | 0.73 | 0.32 | 100 |
| Ours | **24.35** | **0.74** | **0.31** | 90 |

DDRM (Kawar et al., 2022a), DDNM (Wang et al., 2023), and DPS (Chung et al., 2023) for noisy SR, and QGAC (Ehrlich et al., 2020), FBCNN (Jiang et al., 2021), and DDRM-JPEG (Kawar et al., 2022b) for multiple non-align JPEG artifacts correction. DDRM is reported using 20 NFEs in the original paper. For fair comparison, we re-run DDRM for 100 NFEs. We set DDIM inference steps to 100, the inverse strength to 300, $\gamma$ to 0.05, and $M$ to 1. Hence, our method requires 90 NFEs when the degradation model is unknown (30 for DDIM inverse, 30 for DDIM, and 30 for VPS).

For noisy SR, we use $4\times$ average-pooling downsampler and additive Guassian noise with $\sigma = 0.05$. For JPEG artifacts correction, we simulate the real world scenario by multiple non-aligned compression. Specifically, we used cascaded JPEG compression with QF $= (10, 20, 40)$ whose $8 \times 8$ blocks are shifted by $(0, 3, 6)$ pixels respectively. We show upscaling by the inverse upsampler as a baseline for noisy SR and the compressed image itself as a baseline for JPEG artifacts correction. FBCNN is a supervised method and we use the pre-trained model for inference. For DDRM-JPEG and DDNM, we choose the worst case (QF $= 10$) as the degradation model for inference. We use "Ours" to mark the case without knowing degradation model and "Ours*" to mark the scenario of leveraging the degradation model.

Tables 1 to 4 show quantitative results. We can find that i) for noisy SR, even without knowing degradation model, DreamClean can be effective in promoting image quality (compared with baseline) and sometimes surpass those exploiting degradation model (e.g., SNIPS); ii) for complex JPEG artifacts correction, DreamClean outperforms both supervised (QGAC and FBCNN) and unsupervised methods (DDRM-JPEG and DDNM).

## 3.4 Qualitative Experiments

To verify visual results, we conduct qualitative experiments on the various IR tasks using diffusion models (Ho et al., 2020; Dhariwal & Nichol, 2021), ranging from typical linear degradation (image coloration, super-resolution, deblurring), noisy linear degradation (Poisson noise, SR with Gaussian and Poisson noise), non-linear degradation (multiple non-align JPEG artifacts correction (Jiang et al., 2021)), to complex bad weather degradation (rain, snow, raindrop). Figure 2 present visual results across various IR tasks. We can find that DreamClean can produce visually pleasing results while maintaining rather similarity with the input degraded image.

We are also interested in integrating DreamClean with the advanced Stable Diffusion XL (Podell et al., 2023). As shown in Figure 1 and Appendix A.7, although the input images are severely de-

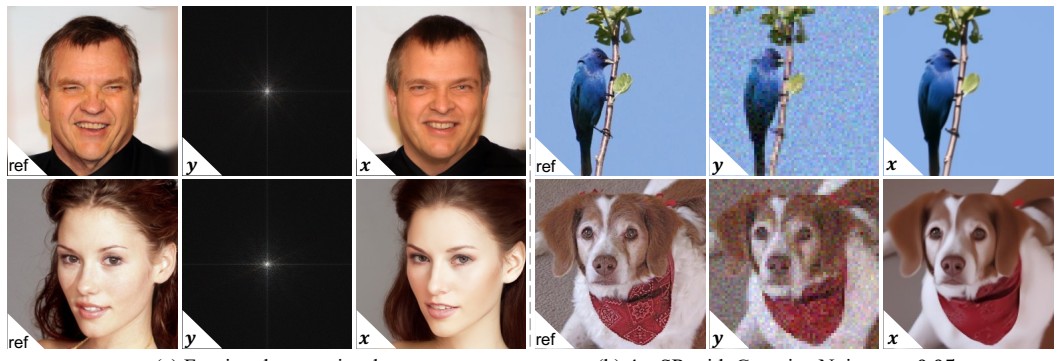

(a) Fourier phase retrieval       (b) 4× SR with Gaussian Noise $\sigma = 0.05$

Figure 7: DreamClean can make use of the degradation model to restore clean images.

Table 5: Ablation study on the $\eta_g$ schedule.

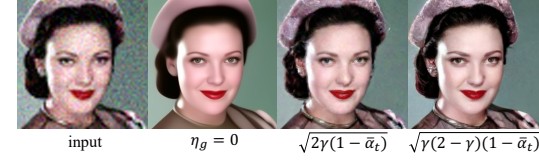

| Schedule | PSNR ↑ | SSIM ↑ | LPIPS ↓ |
|---|---|---|---|
| $0$ | 26.31 | 0.75 | 0.24 |
| $\sqrt{2\gamma(1-\bar{\alpha}_t)}$ | 26.84 | 0.72 | 0.16 |
| $\sqrt{\gamma(2-\gamma)(1-\bar{\alpha}_t)}$ | **27.23** | **0.77** | **0.12** |

Figure 8: Comparison of different $\eta_g$ schedules.

stroyed, DreamClean can still generate high-quality images while keeping similar with input images. Please refer to Appendix A.7 for more visual results.

### 3.5 EXPLOITING DEGRADATION MODEL

Like previous works, DreamClean can also make use of the degradation model to initialize the latents, which is more faithful to the input observation. To validate that, we include experiments on the noisy SR task and challenging Fourier phase retrieval, which aims to restore a clean image based on the magnitude of Fourier transformation of an image. Since degradation model is known, we do not require DDIM inverse to keep similarity and instead resort to Equation (2) for fast inverse. Thus, we only need 60 NFEs. Tables 1 and 2 shows quantitative results and Figure 7 presents the visual results. Although little perceptual information can be found in observations for phase retrieval, DreamClean can utilize the degradation model to recover clean images. Please refer to Appendix A.6 for more quantitative and visual results on other IR tasks, including noisy inpainting, noisy coloration, and noisy deblurring (Uniform and Gaussian kernel).

### 3.6 ABLATION STUDY

We here conduct ablation study on the constraint of $\eta_l$ and $\eta_g$ in Equation (9). Suppose $\eta_l$ still has the form $\eta_l = \gamma(1-\bar{\alpha}_t)$, we investigate two alternate schedules: $\eta_g = 0$ and $\eta_g = \sqrt{2\gamma(1-\bar{\alpha}_t)}$. The former corresponds to plain gradient ascent and the latter corresponds to vanilla lagevien dynamics. We conduct noisy $4\times$ SR on CelebA. Table 5 and Figure 8 present quantitative and visual comparisons. We can find that the schedule of VPS achieves the best score and visual result. It is note that plain gradient ascent cannot produce images. This is because it cannot optimize latents to High Probability Set of a clean image.

## 4 RELATED WORKS

We briefly summarize dominant deep learning approaches for image restoration problems in two categories: supervised and unsupervised methods.

**Supervised Methods.** A deep neural network, which can be CNN (Zhang et al., 2017; Dong et al., 2015; Xia et al., 2023; Ju et al., 2021; Hu et al., 2022), Transformer (Liang et al., 2021; Zamir et al., 2022; Wang et al., 2022; Zhang et al., 2023), Diffusion (Saharia et al., 2022b;a; Whang

et al., 2022) models, etc, is trained to learn to map corrupted images to their clean counterparts under the supervision of a matched degraded-clean dataset (Li et al., 2023). Thanks to powerful representation ability of DNN, supervised methods typically achieve remarkable performance for specific degradation. However, the brilliance comes with a high cost of generality: the performance deteriorates seriously if training samples deviate from the underlying degradation model. Besides, it is also difficult to collect a high-quality dataset if one does not know the true degradation model.

**Unsupervised Methods.** Unsupervised methods bypass the obstacle of matched degraded-clean training pairs by instead exploiting the prior distribution, which can be learned from data or implied in the intrinsic structure of a generator network (Ulyanov et al., 2018; Jagatap & Hegde, 2019). They typically weaken the requirements of matched training pairs to unpaired degraded-clean images (Engin et al., 2018), only ground truth (Venkatakrishnan et al., 2013) or only degraded images (Lehtinen et al., 2018; Bora et al., 2018; Quan et al., 2020; Huang et al., 2021; Mansour & Heckel, 2023). Since the learning is decoupled from specific degradation model, unsupervised methods exhibit high generality (Ongie et al., 2020). They usually utilize the image prior in an iterative procedure. One approach (Venkatakrishnan et al., 2013; Romano et al., 2017; Chang et al., 2017; Sun et al., 2019) is to learn a denoiser from data and apply the denoiser in place of proximal operators in an optimization algorithm, which needs to know the degradation model at test time. Another approach is to learn a generative prior based on training samples using generative adversarial networks (GANs) (Goodfellow et al., 2014). They (Bora et al., 2017; Daras et al., 2021; Pan et al., 2021) optimize the latent input or GAN's weight to minimize the distance between the generated image which is corrupted by the degradation model and input degraded image.

Recently, diffusion models have made significant breakthroughs in image generation. Diffusion models are also widely used to solve various inverse problems in unsupervised way (Choi et al., 2021; Kadkhodaie & Simoncelli, 2021; Kawar et al., 2021; Song et al., 2022; 2021b; Murata et al., 2023). These methods treat a diffusion model as a image prior and exploit desirable property of pre-assumed degradation model. For instance, DDRM (Kawar et al., 2022a;b) tackles with linear inverse problems and perform diffusion in the spectral space, where missing information can be identified and synthesized. Similarly, DDNM (Wang et al., 2023) proposes a zero-shot solver for linear IR problems by refining only the null-space during the reverse diffusion process. DPS (Chung et al., 2023) and GDP (Fei et al., 2023) leverage the degradation model to guide latent variables to ensure consistency with the degraded image for general non-linear degradation. Different from these works, DreamClean significantly weakens the assumption about the degradation model and is capable of producing clean images even without knowing specific form of degradation. Due to its generality, DreamClean can be integrated in the advanced latent diffusion models. In addition, DreamClean inherits the inherent advantage, which avoids training on the matched data, of unsupervised methods. These together constitute the promising prospect of DreamClean.

## 5 LIMITATION AND DISCUSSION

There still remain some limitations. First, although DreamClean can promote visual quality significantly, it can not ensure strict consistency with the input degraded image without knowing degradation model. An effective mechanism to promote consistency deserves further study. Besides, there are some degraded cases that DreamClean struggles to tackle with. For instance, Appendix A.8 provides such a failure case. DreamClean when using diffusion models pre-trained on ImageNet can not remove haze successfully and tends to generate some unexpected content.

## 6 CONCLUSION

We propose a novel unsupervised method named DreamClean for general IR problems. DreamClean figure out a novel avenue to tackle with various degradation types even without supervised training on paired images or assuming specific form of degradation model. DreamClean enjoys elegant theoretical guarantees and achieves remarkable performance across various degradation types, especially for extremely destroyed scenarios. Thanks to its generality, DreamClean also makes it possible to harness the advanced generative models such as Stable Diffusion XL.

ACKNOWLEDGEMENT

This work was supported by the National Natural Science Foundation of China (NSFC) under Grants 62225207 and 62276243.

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

# A APPENDIX

## A.1 PROOF TO THEOREM 2.2

*Proof.* In each iteration of VPS defined by Equation (8), $\boldsymbol{y}_t^{m-1}$ is updated by a gradient term $\eta_l \nabla \log p_t(\boldsymbol{y}_t^{m-1})$ and a noise term $\eta_g \boldsymbol{\epsilon}_g^{m-1}$. Note that the gradient $\nabla \log p_t(\boldsymbol{y}_t)$ for any $\boldsymbol{y}_t \in \mathbb{R}^N$ can be written as

$$\nabla \log p_t(\boldsymbol{y}_t) = \frac{\nabla p_t(\boldsymbol{y}_t)}{p_t(\boldsymbol{y}_t)} \tag{A1}$$

$$= \frac{1}{p_t(\boldsymbol{y}_t)} \int \nabla_{\boldsymbol{y}_t} p_t(\boldsymbol{y}_t|\boldsymbol{x}) p_0(\boldsymbol{x}) \mathrm{d}\boldsymbol{x} \tag{A2}$$

$$= \frac{1}{p_t(\boldsymbol{y}_t)} \int \frac{\sqrt{\bar{\alpha}_t}\boldsymbol{x} - \boldsymbol{y}_t}{1 - \bar{\alpha}_t} p_t(\boldsymbol{y}_t|\boldsymbol{x}) p_0(\boldsymbol{x}) \mathrm{d}\boldsymbol{x} \tag{A3}$$

$$= \frac{1}{1 - \bar{\alpha}_t}(\sqrt{\bar{\alpha}_t}\mathbb{E}[\boldsymbol{x}|\boldsymbol{y}_t] - \boldsymbol{y}_t), \tag{A4}$$

where $p_t(\boldsymbol{y}_t|\boldsymbol{x}) = \mathcal{N}(\boldsymbol{y}_t; \sqrt{\bar{\alpha}_t}\boldsymbol{x}, (1-\bar{\alpha}_t)\mathbf{I})$, $p_0(\boldsymbol{x})$ is the density of clean image distribution, $\mathbb{E}[\boldsymbol{x}|\boldsymbol{y}_t]$ is the expectation of clean image $\boldsymbol{x}$ conditional on $\boldsymbol{y}_t$, and it can be expressed as

$$\mathbb{E}[\boldsymbol{x}|\boldsymbol{y}_t] = \int \boldsymbol{x} \frac{p_t(\boldsymbol{y}_t|\boldsymbol{x})}{p_t(\boldsymbol{y}_t)} p_0(\boldsymbol{x}) \mathrm{d}\boldsymbol{x} \tag{A5}$$

$$= \int \boldsymbol{x} \frac{\mathcal{N}(\boldsymbol{y}_t; \sqrt{\bar{\alpha}_t}\boldsymbol{x}, (1-\bar{\alpha}_t)\mathbf{I})}{\int \mathcal{N}(\boldsymbol{y}_t; \sqrt{\bar{\alpha}_t}\boldsymbol{x}', (1-\bar{\alpha}_t)\mathbf{I}) p_0(\boldsymbol{x}') \mathrm{d}\boldsymbol{x}'} p_0(\boldsymbol{x}) \mathrm{d}\boldsymbol{x} \tag{A6}$$

$$= \int \boldsymbol{x} \frac{\exp\left(-\frac{1}{2(1-\bar{\alpha}_t)}\|\boldsymbol{y}_t - \sqrt{\bar{\alpha}_t}\boldsymbol{x}\|_2^2\right)}{\int \exp\left(-\frac{1}{2(1-\bar{\alpha}_t)}\|\boldsymbol{y}_t - \sqrt{\bar{\alpha}_t}\boldsymbol{x}'\|_2^2\right) p_0(\boldsymbol{x}') \mathrm{d}\boldsymbol{x}'} p_0(\boldsymbol{x}) \mathrm{d}\boldsymbol{x}. \tag{A7}$$

Suppose $\boldsymbol{y}_t$ is a combination of an image $\boldsymbol{x}_0$ and a Gaussian noise $\boldsymbol{\epsilon}$ in the form of $\boldsymbol{y}_t = \sqrt{\bar{\alpha}_t}\boldsymbol{x}_0 + \sqrt{1-\bar{\alpha}_t}\boldsymbol{\epsilon}$, then $\|\boldsymbol{y}_t - \sqrt{\bar{\alpha}_t}\boldsymbol{x}\|_2^2$ in Equation (A7) can be approximated as

$$\|\boldsymbol{y}_t - \sqrt{\bar{\alpha}_t}\boldsymbol{x}\|_2^2 = \bar{\alpha}_t\|\boldsymbol{x}_0 - \boldsymbol{x}\|_2^2 + (1-\bar{\alpha}_t)\|\boldsymbol{\epsilon}\|_2^2 + 2\sqrt{\bar{\alpha}_t(1-\bar{\alpha}_t)}\boldsymbol{\epsilon} \cdot (\boldsymbol{x}_0 - \boldsymbol{x}) \tag{A8}$$

$$\approx \bar{\alpha}_t\|\boldsymbol{x}_0 - \boldsymbol{x}\|_2^2 + (1-\bar{\alpha}_t)\|\boldsymbol{\epsilon}\|_2^2, \tag{A9}$$

where we reasonably assume that the noise term $\boldsymbol{\epsilon}$ is approximately orthogonal to $\boldsymbol{x}_0 - \boldsymbol{x}$. Similarly, we have

$$\|\boldsymbol{y}_t - \sqrt{\bar{\alpha}_t}\boldsymbol{x}'\|_2^2 \approx \bar{\alpha}_t\|\boldsymbol{x}_0 - \boldsymbol{x}'\|_2^2 + (1-\bar{\alpha}_t)\|\boldsymbol{\epsilon}\|_2^2. \tag{A10}$$

Substitute Equations (A9) and (A10) into Equation (A7), we can get an approximation of $\mathbb{E}[\boldsymbol{x}|\boldsymbol{y}_t]$ as

$$\mathbb{E}[\boldsymbol{x}|\boldsymbol{y}_t] \approx \int \boldsymbol{x} \frac{\exp\left(-\frac{1}{2(1-\bar{\alpha}_t)}\left(\bar{\alpha}_t\|\boldsymbol{x}_0 - \boldsymbol{x}\|_2^2 + (1-\bar{\alpha}_t)\|\boldsymbol{\epsilon}\|_2^2\right)\right)}{\int \exp\left(-\frac{1}{2(1-\bar{\alpha}_t)}\left(\bar{\alpha}_t\|\boldsymbol{x}_0 - \boldsymbol{x}'\|_2^2 + (1-\bar{\alpha}_t)\|\boldsymbol{\epsilon}\|_2^2\right)\right) p_0(\boldsymbol{x}') \mathrm{d}\boldsymbol{x}'} p_0(\boldsymbol{x}) \mathrm{d}\boldsymbol{x} \tag{A11}$$

$$= \int \boldsymbol{x} \frac{\exp\left(-\frac{\bar{\alpha}_t}{2(1-\bar{\alpha}_t)}\|\boldsymbol{x}_0 - \boldsymbol{x}\|_2^2\right)}{\int \exp\left(-\frac{\bar{\alpha}_t}{2(1-\bar{\alpha}_t)}\|\boldsymbol{x}_0 - \boldsymbol{x}'\|_2^2\right) p_0(\boldsymbol{x}') \mathrm{d}\boldsymbol{x}'} p_0(\boldsymbol{x}) \mathrm{d}\boldsymbol{x} \tag{A12}$$

$$= \int \boldsymbol{x} \frac{\mathcal{N}(\boldsymbol{x}_0; \boldsymbol{x}, \frac{1-\bar{\alpha}_t}{\bar{\alpha}_t}\mathbf{I})}{\int \mathcal{N}(\boldsymbol{x}_0; \boldsymbol{x}', \frac{1-\bar{\alpha}_t}{\bar{\alpha}_t}\mathbf{I}) p_0(\boldsymbol{x}') \mathrm{d}\boldsymbol{x}'} p_0(\boldsymbol{x}) \mathrm{d}\boldsymbol{x} \tag{A13}$$

$$= \frac{1-\bar{\alpha}_t}{\bar{\alpha}_t} \nabla_{\boldsymbol{x}_0} \log r_t(\boldsymbol{x}_0) + \boldsymbol{x}_0, \tag{A14}$$

where

$$r_t(\boldsymbol{x}_0) \triangleq \int \mathcal{N}(\boldsymbol{x}_0; \boldsymbol{x}, \frac{1-\bar{\alpha}_t}{\bar{\alpha}_t}\mathbf{I}) p_0(\boldsymbol{x}) \mathrm{d}\boldsymbol{x}. \tag{A15}$$

From Equation (A14), it is clear that $\mathbb{E}[\boldsymbol{x}|\boldsymbol{y}_t]$ is approximately only dependent of the image component $\boldsymbol{x}_0$ in $\boldsymbol{y}_t$.

Thus, we can get an approximation of $\nabla \log p_t(\boldsymbol{y}_t)$ by substituting $\boldsymbol{y}_t = \sqrt{\bar{\alpha}_t}\boldsymbol{x}_0 + \sqrt{1-\bar{\alpha}_t}\boldsymbol{\epsilon}$ and Equation (A14) into Equation (A4)

$$\nabla \log p_t(\boldsymbol{y}_t) \approx \frac{\nabla_{\boldsymbol{x}_0} \log r_t(\boldsymbol{x}_0)}{\sqrt{\bar{\alpha}_t}} - \frac{\boldsymbol{\epsilon}}{\sqrt{1-\bar{\alpha}_t}}. \tag{A16}$$

With Equation (A16) and the relation $\boldsymbol{y}_t^{m-1} = \sqrt{\bar{\alpha}_t}\boldsymbol{x}_0^{m-1} + \sqrt{1-\bar{\alpha}_t}\boldsymbol{\epsilon}^{m-1}$, we can obtain $\boldsymbol{y}_t^m$ from the updating rule Equation (8) as

$$\boldsymbol{y}_t^m \approx \sqrt{\bar{\alpha}_t}\left(\boldsymbol{x}_0^{m-1} + \frac{\eta_l}{\bar{\alpha}_t}\nabla \log r_t(\boldsymbol{x}_0^{m-1})\right) + (\sqrt{1-\bar{\alpha}_t} - \frac{\eta_l}{\sqrt{1-\bar{\alpha}_t}})\boldsymbol{\epsilon}^{m-1} + \eta_g\boldsymbol{\epsilon}_g^m \tag{A17}$$

$$= \sqrt{\bar{\alpha}_t}\underbrace{\left(\boldsymbol{x}_0^{m-1} + \frac{\eta_l}{\bar{\alpha}_t}\nabla \log r_t(\boldsymbol{x}_0^{m-1})\right)}_{\boldsymbol{x}_0^m} + \sqrt{1-\bar{\alpha}_t}\boldsymbol{\epsilon}^m, \tag{A18}$$

where the equality holds because the linear combination of two independent Gaussian noises is still Gaussian, and their variances satisfies

$$1 - \bar{\alpha}_t = (\sqrt{1-\bar{\alpha}_t} - \frac{\eta_l}{\sqrt{1-\bar{\alpha}_t}})^2 + \eta_g^2, \tag{A19}$$

which is guaranteed by the relationship of $\eta_l$ and $\eta_g$ defined by Equation (9).

Compare Equation (A18) with $\boldsymbol{y}_t^{m-1} = \sqrt{\bar{\alpha}_t}\boldsymbol{x}_0^{m-1} + \sqrt{1-\bar{\alpha}_t}\boldsymbol{\epsilon}^{m-1}$, we can find that the variance of the noise component keeps unchanged.

As for the image component $\boldsymbol{x}_0^m$, the updating rule $\boldsymbol{x}_0^m = \boldsymbol{x}_0^{m-1} + \frac{\eta_l}{\bar{\alpha}_t}\nabla \log r_t(\boldsymbol{x}_0^{m-1})$ is exactly the gradient ascent, with the optimization target $\log r_t(\boldsymbol{x}_0)$ and step size $\frac{\eta_l}{\bar{\alpha}_t}$. Thus, $\boldsymbol{x}_0^m$ will converge to a local maximum of $\log r_t(\boldsymbol{x}_0)$, denoted as $\boldsymbol{x}_0^*$, that satisfies

$$\nabla \log r_t(\boldsymbol{x}_0^*) = \boldsymbol{0}. \tag{A20}$$

Note that $r_t(\boldsymbol{x}_0)$ represents the distribution induced by adding a Gaussian noise with variance $\frac{1-\bar{\alpha}_t}{\bar{\alpha}_t}$ to the clean image distribution $p_0$, as defined by Equation (A15). Thus, for small $t$ which corresponds to small variance, the maxima of $\log r_t(\boldsymbol{x}_0)$ will coincide with clean images, under the mixture of Dirac assumption on image distribution.

Finally, we can conclude that $\boldsymbol{y}_t^M$ will get into $\mathcal{T}_t^N(\boldsymbol{x}_0^*; |\delta_\pm|)$ for some small $\delta_\pm$ and sufficient large $M$ by verifying that

$$-\frac{1}{N}\log p_t(\boldsymbol{y}_t^M|\boldsymbol{x}_0^*) \tag{A21}$$

$$= -\frac{1}{N}\log \frac{1}{(2\pi(1-\bar{\alpha}_t))^{N/2}}\exp\left(-\frac{1}{2(1-\bar{\alpha}_t)}\|\sqrt{\bar{\alpha}_t}\boldsymbol{x}_0^M + \sqrt{1-\bar{\alpha}_t}\boldsymbol{\epsilon}^M - \sqrt{\bar{\alpha}_t}\boldsymbol{x}_0^*\|_2^2\right) \tag{A22}$$

$$= \frac{1}{2}\log 2\pi(1-\bar{\alpha}_t) + \frac{1}{2N(1-\bar{\alpha}_t)}\|\sqrt{\bar{\alpha}_t}(\boldsymbol{x}_0^M - \boldsymbol{x}_0^*) + \sqrt{1-\bar{\alpha}_t}\boldsymbol{\epsilon}^M\|_2^2 \tag{A23}$$

$$\to \frac{1}{2}\log 2\pi(1-\bar{\alpha}_t) + \frac{1}{2} + \delta_\pm, \tag{A24}$$

where that last limitation holds because $\boldsymbol{x}_0^M \to \boldsymbol{x}_0^*$ as $M$ increases, $\delta_\pm = \frac{1}{2}(\frac{\|\boldsymbol{\epsilon}^M\|}{N} - 1)$ is small because $\frac{\|\boldsymbol{\epsilon}^M\|}{N} \approx 1$ for large $N$, which is guaranteed by the high dimensionality of images and the law of large numbers. □

## A.2 THE PROPERTY OF HIGH PROBABILITY SET

We define a new variable $v_{t,i} = x_{t,i} - \sqrt{\bar{\alpha}_t}x_{0,i}$. Since elements of $\boldsymbol{x}_t$ are conditional independent given $\boldsymbol{x}_0$, $\{v_{t,i}\}$ are independent and identically distributed and $v_{t,i} \sim \mathcal{N}(0; 1-\bar{\alpha}_t)$. Suppose $H$ is Shannon entropy of $\mathcal{N}(0; 1-\bar{\alpha}_t)$, according to weak law of large numbers, we have

$$-\frac{1}{N}\log p_t(\boldsymbol{x}_t|\boldsymbol{x}_0) = -\frac{1}{N}\log p_t(x_{t,0}, x_{t,1}, \ldots x_{t,N}|x_{t,0}, x_{t,1}, \ldots x_{t,N}) \tag{A25}$$

$$= -\frac{1}{N}\log p_t(v_{t,0}, v_{t,1}, \ldots v_{t,N}) \tag{A26}$$

$$\to H \quad \text{in probability} \tag{A27}$$

for sufficiently large $N$. For a high-resolution image, $N$ is typically large. Therefore, High Probability Set contains most of probability.

## A.3 EXTENSION TO REAL-WORLD APPLICATION

We extend DreamClean to more complex real-world applications. First, we perform real-world image denoising on SIDD dataset (Abdelhamed et al., 2018). For quantitative comparison, we test original unprocessed data scores (Baseline) and evaluate the classic BM3D (Dabov et al., 2007) as comparison. Table A1 reveals that DreamClean can effectively tackle with real-world noise (+8.26 dB compared with Baseline). Moreover, Figure A1 qualitatively presents results of DreamClean on more applications, including restoring real-world bad weather corrupted images, old photo restoration, and real-world image denoising.

Table A1: Quantitative results on SIDD.

| SIDD | PSNR↑ | SSIM↑ | LPIPS↓ |
|------|-------|-------|--------|
| Baseline | 23.66 | 0.35 | 0.58 |
| BM3D | 25.65 | 0.68 | N/A |
| Ours | 31.92 | 0.76 | 0.23 |

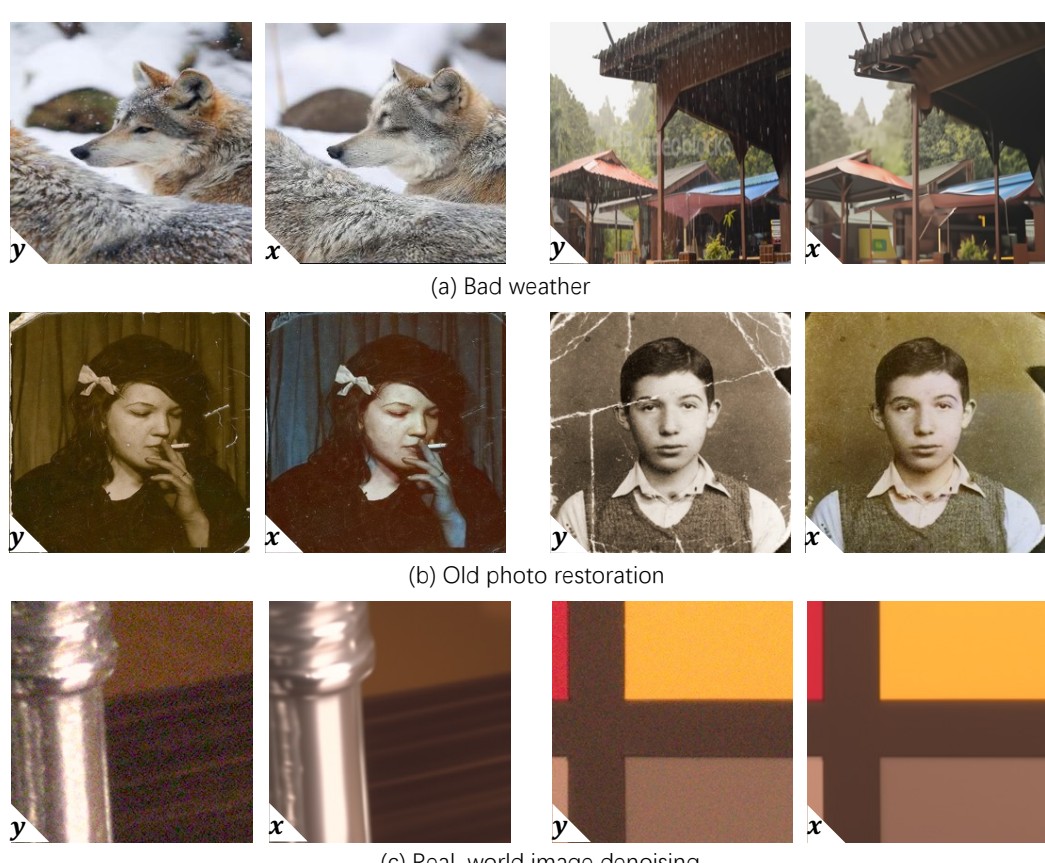

(a) Bad weather

(b) Old photo restoration

(c) Real-world image denoising

Figure A1: Extension to real-world applications. $\boldsymbol{y}$: the degraded image, $\boldsymbol{x}$: our result.

## A.4 VISUALIZATION OF DDIM AND VPS

The visualization in Figure A2 intuitively illustrates the respective function of the VPS and DDIM step respectively. We can find that after VPS correction, the original degraded artifatcs translate

to Gaussian-like noise. Therefore, VPS step is responsiable for correcting the corrupted low-probability latents. Moreover, after DDIM step, the amount of noise is decreased progressively. Thus, DDIM step is responsible for progressively reducing the amount of Gaussian noise contained in latents.

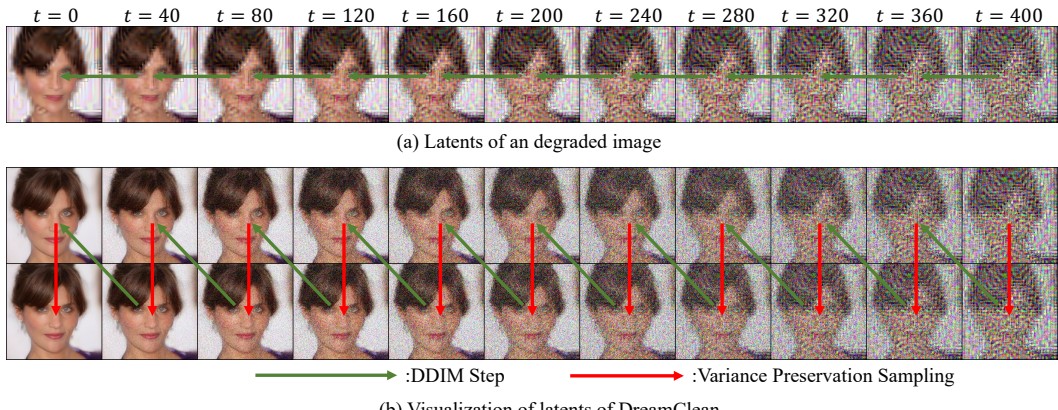

(a) Latents of an degraded image

(b) Visualization of latents of DreamClean

Figure A2: Visualization of latents of DDIM and VPS. VPS translates original degraded artifacts to Gaussian-like noise and DDIM step is responsible for progressively reducing the amount of Gaussian noise contained in latents.

## A.5 ALGORITHM

We present DDIM inversion in Algorithm A1, Variance Preservation Sampling algorithm in Algorithm A2 and DreamClean algorithm in Algorithm A3.

---

**Algorithm A1** DDIM Inversion

---

**Require:** $\boldsymbol{y}, \boldsymbol{y}_\tau$
**Require:** a pre-trained diffusion model $\boldsymbol{\epsilon}_\theta$
1: $\boldsymbol{y}_0 \leftarrow \boldsymbol{y}$
2: **for** $t = 0$ to $\tau - 1$ **do**
3: $\quad \boldsymbol{y}_{t+1} \leftarrow \sqrt{\bar{\alpha}_{t+1}} \left( \frac{\boldsymbol{y}_t - \sqrt{1-\bar{\alpha}_t} \boldsymbol{\epsilon}_\theta(\boldsymbol{y}_t, t)}{\sqrt{\bar{\alpha}_t}} \right) + \sqrt{1 - \bar{\alpha}_{t+1}} \boldsymbol{\epsilon}_\theta \left( \boldsymbol{y}_t, t \right)$
4: **end for**
5: **return** $\boldsymbol{y}_\tau$

---

---

**Algorithm A2** Variance Preservation Sampling

---

**Require:** $M, \eta_l, \eta_g, \boldsymbol{y}_t$
**Require:** a pre-trained diffusion model $\boldsymbol{\epsilon}_\theta$
1: $\boldsymbol{y}_t^0 \leftarrow \boldsymbol{y}_t$
2: **for** $m = 0$ to $M - 1$ **do**
3: $\quad \boldsymbol{y}_t^{m+1} \leftarrow \boldsymbol{y}_t^m - \eta_l \frac{\boldsymbol{\epsilon}_\theta(\boldsymbol{y}_t^m, t)}{\sqrt{1-\bar{\alpha}_t}} + \eta_g \boldsymbol{\epsilon}$
4: **end for**
5: **return** $\boldsymbol{y}_t^M$

---

## A.6 EXPLOITING DEGRADATION MODEL

DreamClean is orthogonal to previous works that make use of the degradation model. To validate that, we perform experiments on classic noisy linear tasks including uniform deblurring, deblurring with Gaussian kernel, inpainting and colorization with Gaussian noise $\sigma = 0.05$ on ImageNet

---

**Algorithm A3** DreamClean

---

**Require:** $\boldsymbol{y}, \tau, M, \eta_l, \eta_g$
**Require:** a pre-trained diffusion model $\boldsymbol{\epsilon}_\theta$
 1: $\boldsymbol{y}_0 \leftarrow \boldsymbol{y}$ #DDIM inversion, producing the latent $\boldsymbol{y}_\tau$
 2: **for** $t = 0$ to $\tau - 1$ **do**
 3: $\quad \boldsymbol{y}_{t+1} \leftarrow \sqrt{\bar{\alpha}_{t+1}} \left( \frac{\boldsymbol{y}_t - \sqrt{1-\bar{\alpha}_t}\boldsymbol{\epsilon}_\theta(\boldsymbol{y}_t, t)}{\sqrt{\bar{\alpha}_t}} \right) + \sqrt{1 - \bar{\alpha}_{t+1}}\boldsymbol{\epsilon}_\theta\left(\boldsymbol{y}_t, t\right)$
 4: **end for**
 5: **for** $t = \tau$ to $1$ **do**
 6: $\quad \boldsymbol{y}_t^0 \leftarrow \boldsymbol{y}_t$ #Variance Preservation Sampling, no change to $t$
 7: $\quad$ **for** $m = 0$ to $M - 1$ **do**
 8: $\quad\quad \boldsymbol{y}_t^{m+1} \leftarrow \boldsymbol{y}_t^m - \eta_l \frac{\boldsymbol{\epsilon}_\theta(\boldsymbol{y}_t^m, t)}{\sqrt{1-\bar{\alpha}_t}} + \eta_g \boldsymbol{\epsilon}$
 9: $\quad$ **end for**
10: $\quad \boldsymbol{y}_t \leftarrow \boldsymbol{y}_t^M$ #DDIM Step, from $t$ to $t-1$
11: $\quad \boldsymbol{y}_{t-1} \leftarrow \sqrt{\bar{\alpha}_{t-1}} \left( \frac{\boldsymbol{y}_t - \sqrt{1-\bar{\alpha}_{t-1}}\boldsymbol{\epsilon}_\theta(\boldsymbol{y}_t, t)}{\sqrt{\bar{\alpha}_t}} \right) + \sqrt{1 - \bar{\alpha}_{t-1}}\boldsymbol{\epsilon}_\theta\left(\boldsymbol{y}_t, t\right)$
12: **end for**
13: **return** $\boldsymbol{y}_0$

---

1K (Deng et al., 2009) and CelebA 1K (Karras et al., 2018). We conduct Variance Preservation Sampling on null-space (Wang et al., 2023). Tables A2 and A3 present the quantitative results. Figure A10 presents visual results.

Table A2: Quantitative results on ImageNet.

| **ImageNet** Method | Deblurring(uniform) PSNR↑/SSIM↓/LPIPS↓ | Deblurring(gauss) PSNR↑/SSIM↓/LPIPS↓ | Colorization LPIPS↓ | Inpainting PSNR↑/SSIM↓/LPIPS↓ |
|---|---|---|---|---|
| Baseline | 18.35/0.26/0.87 | 17.79/0.31/0.71 | 0.54 | 12.32/0.46/0.40 |
| DPS | N/A | 24.64/0.67/0.30 | N/A | 22.14/0.73/0.26 |
| DDRM | 25.09/0.71/**0.30** | 27.82/0.80/**0.24** | 0.25 | 23.09/0.83/0.13 |
| DDNM | 24.28/0.65/0.40 | 26.43/0.75/0.29 | 0.33 | 23.12/0.82/0.13 |
| Ours | **26.78/0.75**/0.33 | **28.92/0.82**/0.24 | **0.11** | **23.18/0.83/0.11** |

Table A3: Quantitative results on CelebA.

| **CelebA** Method | Deblurring(uniform) PSNR↑/SSIM↓/LPIPS↓ | Deblurring(gauss) PSNR↑/SSIM↓/LPIPS↓ | Colorization LPIPS↓ | Inpainting PSNR↑/SSIM↓/LPIPS↓ |
|---|---|---|---|---|
| Baseline | 19.21/0.31/0.86 | 18.06/0.34/0.73 | 0.61 | 12.18/0.40/0.42 |
| DPS | N/A | 28.83/0.81/0.11 | N/A | 22.72/0.82/0.13 |
| DDRM | 28.06/0.80/**0.13** | 30.52/0.85/**0.08** | 0.13 | 23.24/0.85/0.09 |
| DDNM | 28.98/0.82/0.14 | 30.37/0.85/0.11 | 0.13 | 23.23/0.85/0.09 |
| Ours | **31.17/0.88/0.13** | **32.71/0.91**/0.09 | **0.10** | **24.71/0.87/0.07** |

## A.7 MORE VISUAL RESULTS

We present more visual results on various IR tasks using diffusion models (Ho et al., 2020; Dhariwal & Nichol, 2021) in Figure A9 as well as the Stable Diffusion XL (Podell et al., 2023) in Figures A7 and A8. DreamClean exhibits strong robustness about degradation types and compatibility with diffusion models.

## A.8 FAILURE CASE

We present a failure case of in Figure A3. DreamClean  fai ls to remove haze and tends to generate unexpected results using diffusion models (Dhariwal & Nichol, 2021) pre-trained on ImageNet.

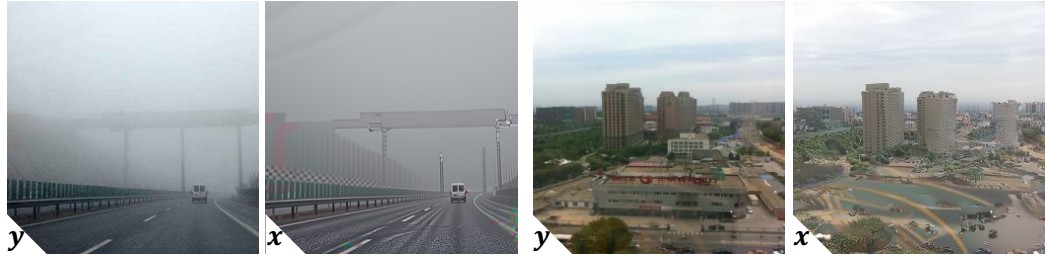

Figure A3: Failure case of our method.

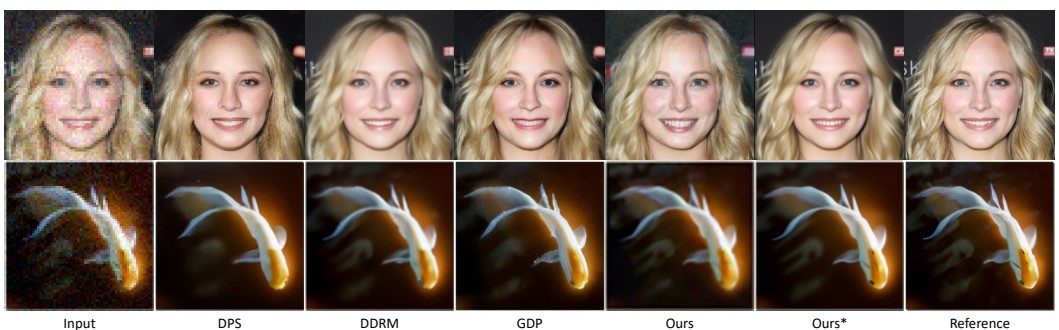

Figure A4: Visual comparison of $4\times$ SR with $\sigma = 0.05$.

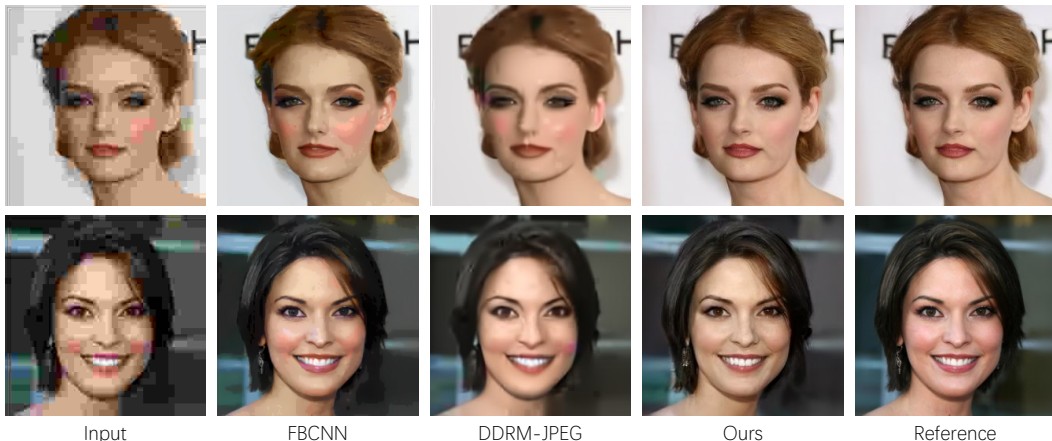

Figure A5: Visual comparison of JPEG artifacts correction.

## A.9 ON THE ACCUMULATION OF VPS CORRECTING EFFECT

From the previous deduction in Appendix A.1, we can find that

$$\boldsymbol{y}_t^1 \approx \sqrt{\bar{\alpha}_t} \underbrace{\left(\boldsymbol{x}_0 + \frac{\eta_l}{\bar{\alpha}_t} \nabla \log r_t(\boldsymbol{x}_0)\right)}_{\boldsymbol{x}_0^1} + \sqrt{1-\bar{\alpha}_t}\boldsymbol{\epsilon}_t, \tag{A28}$$

The denoising process can be written as

$$\boldsymbol{y}_{t-1} = \frac{\sqrt{\bar{\alpha}_{t-1}}}{\sqrt{\bar{\alpha}_t}}\left(\boldsymbol{y}_t^1 - \frac{1-\frac{\bar{\alpha}_t}{\bar{\alpha}_{t-1}}}{\sqrt{1-\bar{\alpha}_t}}\boldsymbol{\epsilon}_{\boldsymbol{\theta}}(\boldsymbol{y}_t^1, t)\right) + \sigma_t \boldsymbol{\epsilon}_{t-1} \tag{A29}$$

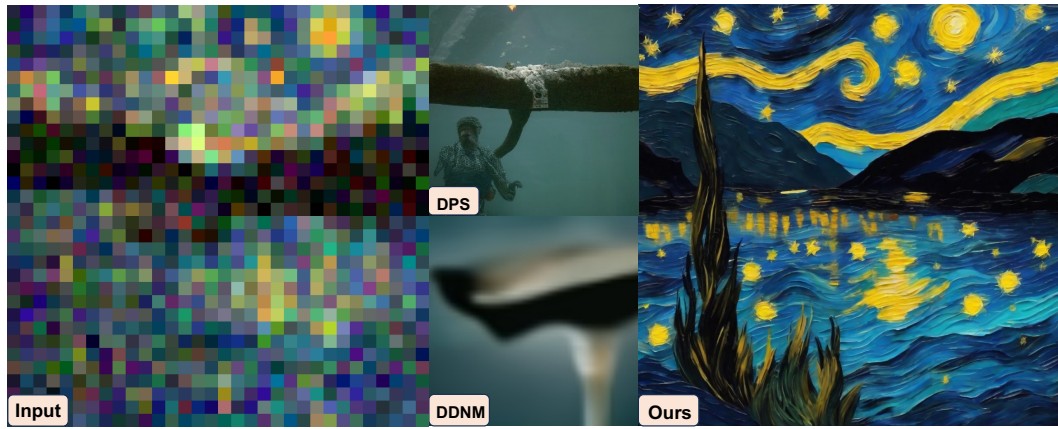

Figure A6: Visual comparison of $32\times$ SR with $\sigma = 0.1$ using Stable Diffusion XL. DDNM and DPS generate unrelated content because they use guided-diffusion pretrained on ImageNet.

The denosing network $\epsilon_{\theta}(\boldsymbol{y}_t^1, t)$, by definition, will predict the noise term of $\boldsymbol{y}_t^1$. So we have

$$\epsilon_{\theta}(\boldsymbol{y}_t^1, t) \approx \epsilon_t, \tag{A30}$$

$$\boldsymbol{y}_{t-1} \approx \sqrt{\bar{\alpha}_{t-1}}\left(\boldsymbol{x}_0 + \frac{\eta_l}{\bar{\alpha}_t}\nabla\log r_t(\boldsymbol{x}_0)\right) - \frac{\sqrt{\bar{\alpha}_{t-1}/\bar{\alpha}_t} - \sqrt{\bar{\alpha}_t/\bar{\alpha}_{t-1}}}{\sqrt{1-\bar{\alpha}_t}}\epsilon_t \tag{A31}$$

$$+ \sqrt{\frac{\bar{\alpha}_{t-1}}{\bar{\alpha}_t}}\epsilon_t + \sigma_t\epsilon_{t-1} \tag{A32}$$

$$= \sqrt{\bar{\alpha}_{t-1}}\left(\boldsymbol{x}_0 + \frac{\eta_l}{\bar{\alpha}_t}\nabla\log r_t(\boldsymbol{x}_0)\right) + \sqrt{\frac{\bar{\alpha}_t}{\bar{\alpha}_{t-1}}}\frac{1-\bar{\alpha}_{t-1}}{\sqrt{1-\bar{\alpha}_t}}\epsilon_t + \sigma_t\epsilon_{t-1}. \tag{A33}$$

Now if we take $\sigma_t = \sqrt{\frac{1-\bar{\alpha}_{t-1}}{1-\bar{\alpha}_t}(1-\frac{\bar{\alpha}_t}{\bar{\alpha}_{t-1}})}$, which is the most popular setting, we can find the final noise strength is

$$s^2 = \frac{\bar{\alpha}_t}{\bar{\alpha}_{t-1}}\frac{(1-\bar{\alpha}_{t-1})^2}{1-\bar{\alpha}_t} + \frac{1-\bar{\alpha}_{t-1}}{1-\bar{\alpha}_t}(1-\frac{\bar{\alpha}_t}{\bar{\alpha}_{t-1}}) = 1 - \bar{\alpha}_{t-1} \tag{A34}$$

So we can find after a denoising sampling step following the Variance Preservation Sampling, we have

$$\boldsymbol{y}_{t-1} = \sqrt{\bar{\alpha}_{t-1}}\underbrace{\left(\boldsymbol{x}_0 + \frac{\eta_l}{\bar{\alpha}_t}\nabla\log r_t(\boldsymbol{x}_0)\right)}_{\boldsymbol{x}_0(t)} + \sqrt{1-\bar{\alpha}_{t-1}}\epsilon_{t-1} \tag{A35}$$

$$= \sqrt{\bar{\alpha}_{t-1}}\boldsymbol{x}_0(t) + \sqrt{1-\bar{\alpha}_{t-1}}\epsilon_{t-1}. \tag{A36}$$

We can conclude that during the Variance Preservation Sampling and denosing sampling, the $\boldsymbol{y}_t$ sequence is actually updating an $\boldsymbol{x}_0$ prediction results. In each step, VPS corrects the corrupted $\boldsymbol{x}_0(t)$ component by the gradient term $\frac{\eta_l}{\bar{\alpha}_t}\nabla\log r_t(\boldsymbol{x}_0)$, while the denoising step preserves the correcting effect. Thus, the correcting effect of VPS can be accumulated along the sampling process.

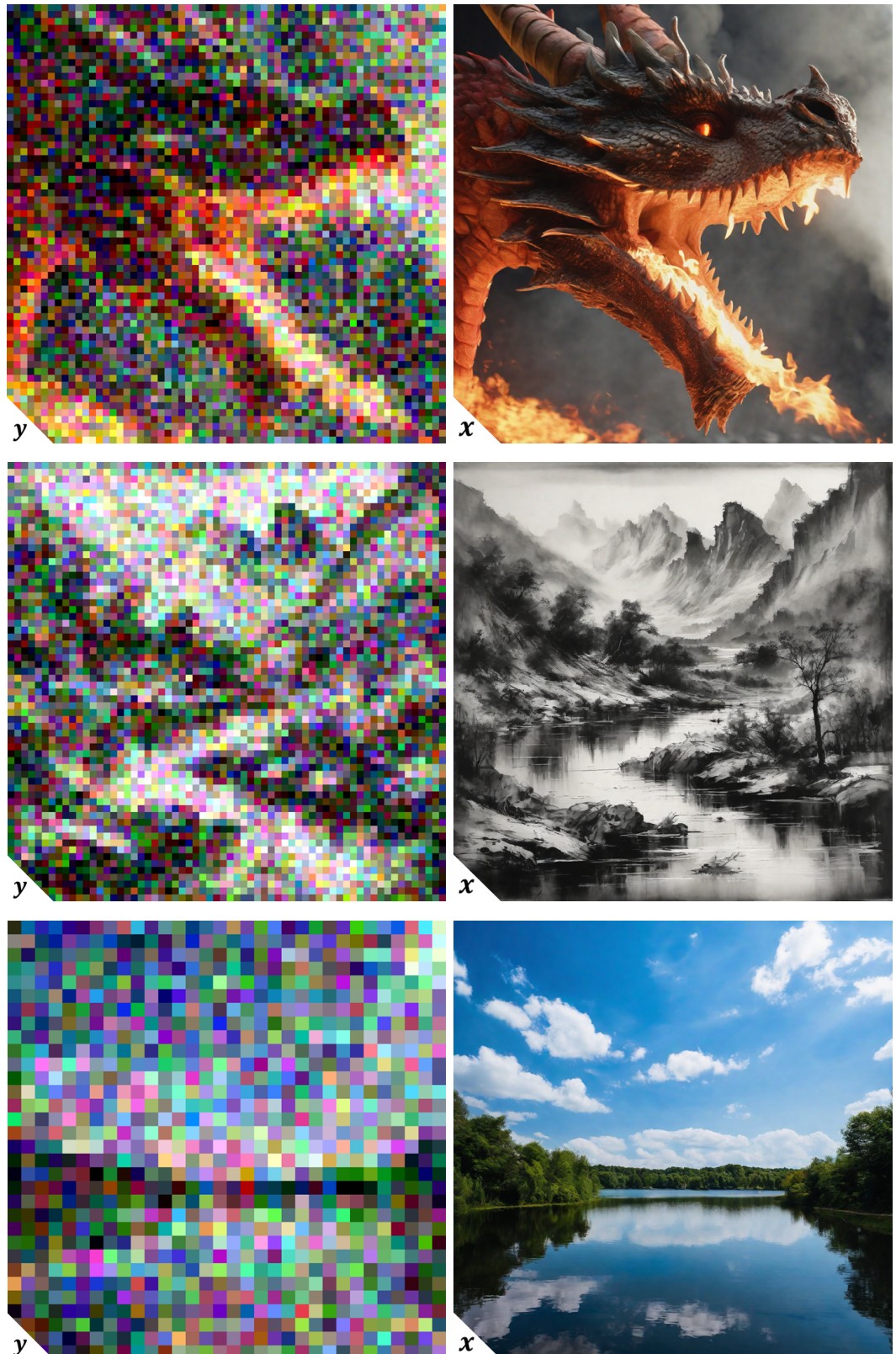

Figure A7: Noisy SR results of DreamClean using Stable Diffusion XL. The image resolution is $1024 \times 1024$. $\boldsymbol{y}$: the degraded image, $\boldsymbol{x}$: our result.

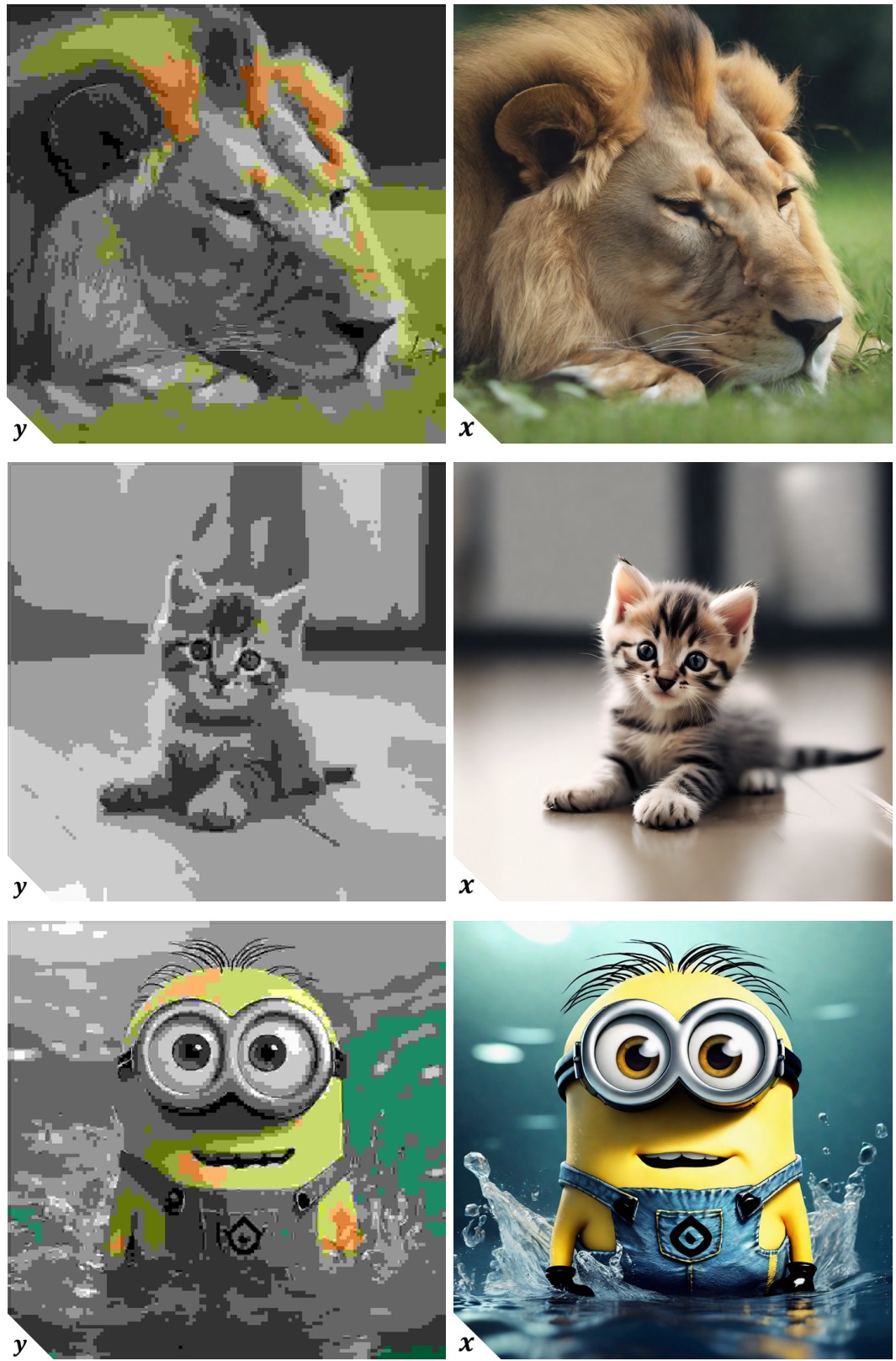

Figure A8: JPEG artifacts correction of DreamClean using Stable Diffusion XL. The image resolution is $1024 \times 1024$. $\boldsymbol{y}$: the degraded image, $\boldsymbol{x}$: our result.

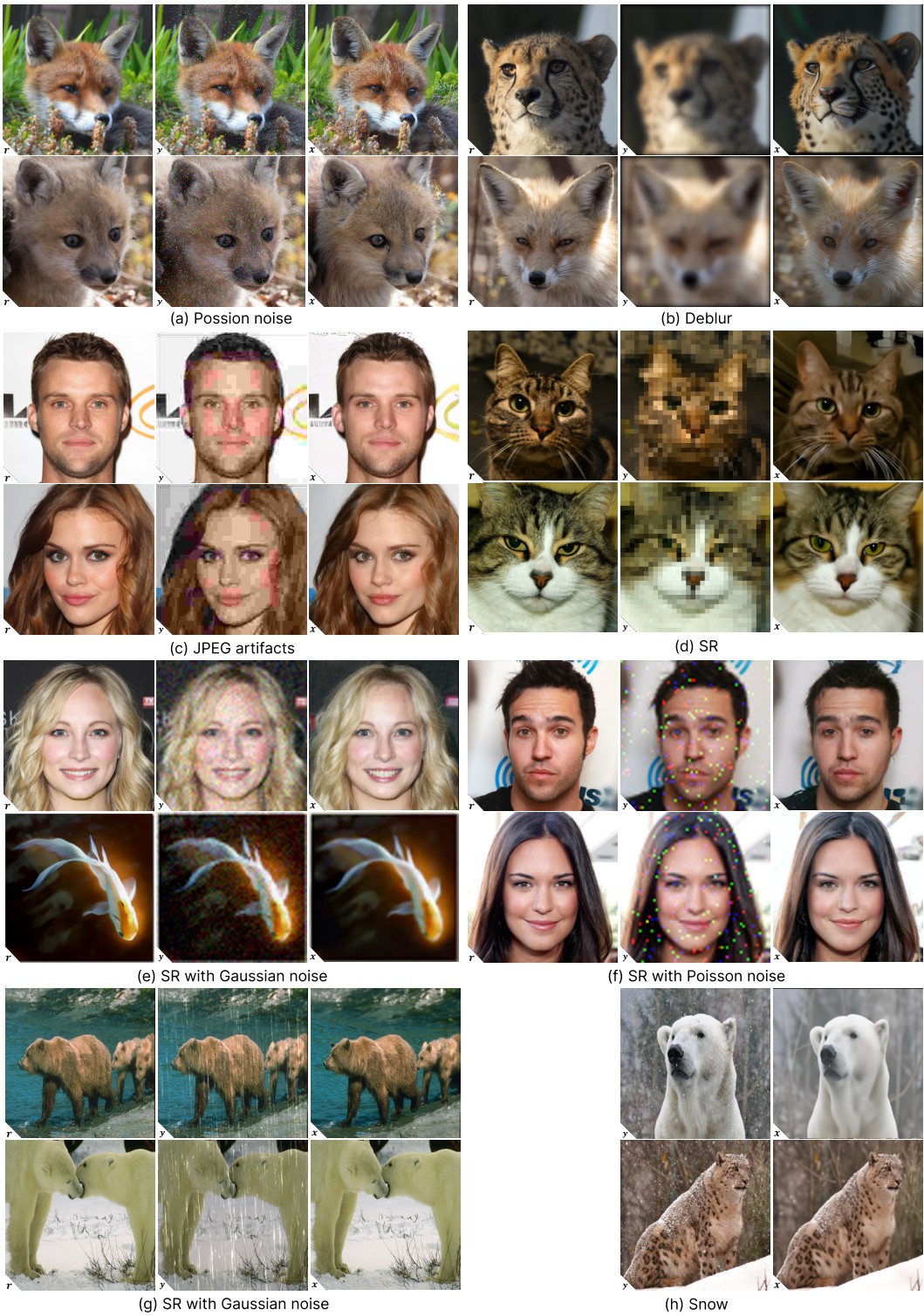

Figure A9: DreamClean can tackle with linear degradation, noisy linear degradation, non-linear degradation and complex bad weather degradation in a blind way. $r$: the reference image, $y$: the degraded image, and $x$: our result.

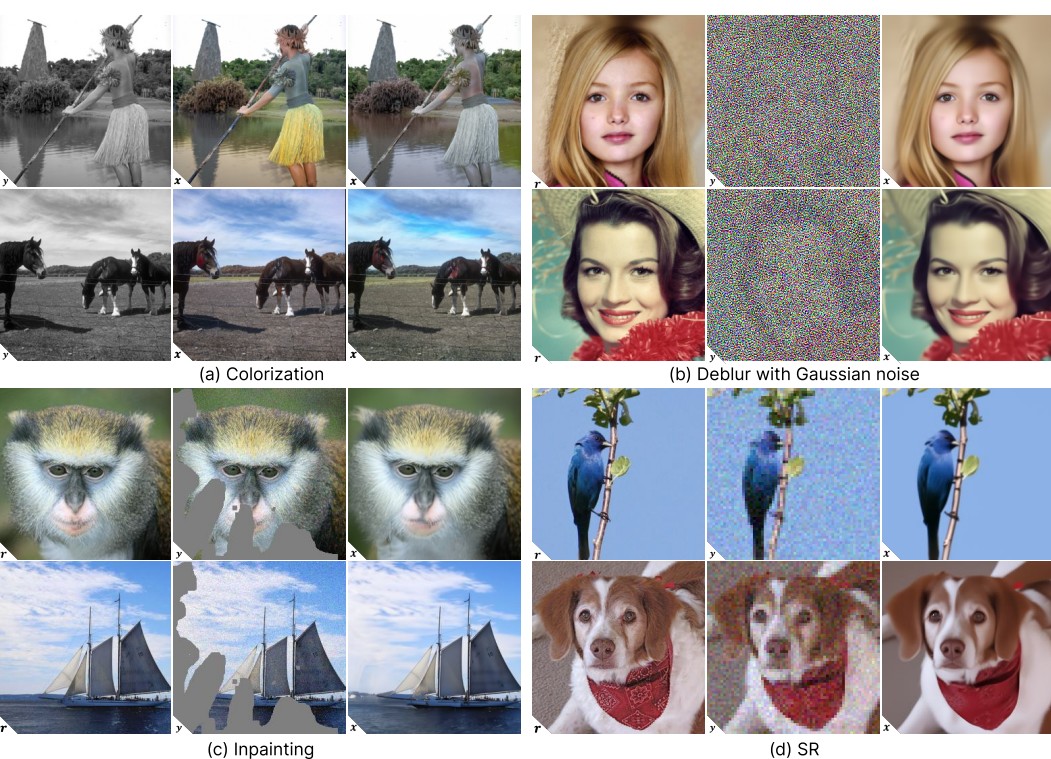

(a) Colorization             (b) Deblur with Gaussian noise

(c) Inpainting             (d) SR

Figure A10: DreamClean can make use of the degradation model to reconstruct clean images. $r$: the reference image, $y$: the degraded image, $x$: our result.

