# OpenReview forum: "DreamClean: Restoring Clean Image Using Deep Diffusion Prior"
_ICLR.cc/2024/Conference — ICLR 2024 poster_

### Official Review · Reviewer_6gxh · 2023-10-26

**Soundness:** 3 good
**Presentation:** 3 good
**Contribution:** 4 excellent
**Rating:** 8
**Confidence:** 3

**Summary:**

The paper introduces a novel generic image restoration (IR) approach called DreamClean that recovers degraded images back to high-quality reconstructed images without heavily relying on or knowing degraded models. The DreamClean method consists of two essential steps to restore corrupt images (e.g., noisy images, low-resolution images, or artifacts generated by JPEG compression)to the reconstructed images. First, the method utilizes a pre-trained diffusion model to encode the corrupt image images to latent embeddings in the diffusion models. The paper then presents a novel sampling strategy, Variance Preservation Sampling (VPS), mimicking a process of generating clean images to guide low-probability latent toward nearby high-probability regions for high-quality image synthesis. The contribution of this paper is threefold: 1. the paper introduces a novel aspect of restoring clean images without knowing prior degraded model information. The proposed method could be an IR technique for image restoration applications with variations of camera devices. 2. the proposed method could be generalized to diverse degraded images without training multiple neural network models. 3. the paper evaluates the proposed method on multiple datasets and types of degraded images. The experiment shows DreamClean achieves competitive quantitative results compared with other IR methods.

**Strengths:**

The strengths of this paper are as follows:

1. This paper proposes a generic IR method that can tackle corrupt images generated by variations of degraded models without training multiple network models. The paper provides theoretical and empirical proof to demonstrate the DreamClean method can synthesize high-quality clean images given the corrupt images.

2. The paper proposed a novel sampling strategy, the VPS method, in diffusion-based models to improve the chance of yielding clean images. The paper demonstrates the capability of VPS through theoretical proof and empirical results. The experimentation shows that DeamClean achieves higher quantitative results under different image quality metrics.

3. This paper contains comprehensive quantitative results on image quality comparisons for distinct IR methods. The DreamClean method outperforms other methods on most degraded models on different datasets.

**Weaknesses:**

This paper contains two weaknesses in terms of the proposed method:

1. The DreamClean method can synthesize clean images given the corrupt images; however, the proposed method does not consider consistency between the corrupt images and the reconstructed clean images due to a lack of prior degraded models. The proposed method can fail some restored images by synthesizing unexpected clean images.

2. The proposed method utilizes the pre-trained diffusion model on a particular large-scale dataset (i.e., ImageNet) to embed the corrupt images to the latent embeddings. Then, the method applies VPS to sample clean images during the sampling phase in diffusion models. Due to using the pre-trained model, the proposed method can generate unexpected images.

**Questions:**

There is a question I would like to ask to clarify all doubts in the paper:

1. In section 3.3, which is quantitative experiments, the paper uses 100 NFEs rather than a configuration in the original paper for DDRM. Does it affect the quantitative comparisons because 100 NFEs may not be the configuration for DDRM?

---

> ### Author Response · Authors · 2023-11-19
> **Response to Reviewer 6gxh**
>
> **Q:** The DreamClean method can synthesize clean images given the corrupt images; however, the proposed method does not consider consistency between the corrupt images and the reconstructed clean images due to a lack of prior degraded models. The proposed method can fail some restored images by synthesizing unexpected clean images.
>
> **A:** We agree that the consistency with corrupt images is important for IR problem. In this work, we also consider the consistency issue carefully but under much more challenging setting: the degradation model is unknown. Based on the fact that ODE sampling is approximately invertible, the inverted latents should contain desirable information about the input image. Hence, we leverage DDIM inversion to achieve approximate consistency. We admit that ODE inversion cannot ensure strict consistency with the input image without knowing the degradation model. It is an open problem and we are excited about the coming future that in-depth reseaches that can solve this problem thoroughly or mostly.
> Besides, orthogonal to previous methods, when the degradation model is known, DreamClean can also accomplish more faithful image restoration.
>
> **Q:** The proposed method utilizes the pre-trained diffusion model on a particular large-scale dataset (i.e., ImageNet) to embed the corrupt images to the latent embeddings. Then, the method applies VPS to sample clean images during the sampling phase in diffusion models. Due to using the pre-trained model, the proposed method can generate unexpected images.
>
> **A:** The concern that generating unexpected content because of the pre-trained model is reasonable. Like other generative methods for IR [1], our method also inherits the in-built stochasticity of diffusion models, which benefits diversity but with the risk of yielding undesirable results. As show in limitation (Sec. 5), diffusion models pretrained on ImageNet struggle to generate daily scenes such as altitude view of buildings. We figure out this problem clearly in limitation section (Sec. 5). It is an open problem that deserves further in-depth inverstigations.
>
> **Q:** In section 3.3, which is quantitative experiments, the paper uses 100 NFEs rather than a configuration in the original paper for DDRM. Does it affect the quantitative comparisons because 100 NFEs may not be the configuration for DDRM?
>
> **A:** We use $100$ NFEs for DDRM to keep the comparison in the context of the same or similar computation. In fact, as revealed in Table 1, DDRM with $100$ NFEs produces slightly better results than that with original $20$ NFEs.
>
> **Table 1: DDRM performace with different NFEs.**
> |CelebA|PSNR$\uparrow$|SSIM$\uparrow$|LPIPS$\downarrow$|NFEs$\downarrow$|
> |:----|:----:|:----:|:----:|:----:|
> |DDRM|29.16|0.83|0.11|20|
> |DDRM|29.21|0.83|0.09|100|
>
> |ImageNet|PSNR$\uparrow$|SSIM$\uparrow$|LPIPS$\downarrow$|NFEs$\downarrow$|
> |:----|:----:|:----:|:----:|:----:|
> |DDRM|25.22|0.71|0.32|20|
> |DDRM|25.67|0.73|0.30|100|
>
> [1] Zero-Shot Image Restoration Using Denoising Diffusion Null-Space Model. ICLR 2023.

---

> > ### Comment · Reviewer_6gxh · 2023-11-22
> >
> > Thanks for author's response. The author's response has addressed my concerns.

---

### Official Review · Reviewer_HBur · 2023-10-30

**Soundness:** 2 fair
**Presentation:** 3 good
**Contribution:** 3 good
**Rating:** 6
**Confidence:** 4

**Summary:**

This paper works on unsupervised image restoration using the pre-trained diffusion models as deep diffusion prior. It relaxes the linear degradation model and extends to non-linear and bad weather degradation. The faithfulness of the image restoration is achieved via the rich image prior embedded in the pre-trained models and DDIM inversion. The realness is satisfied through the proposed variance preservation sampling, where the latents are driving to the high probability set. Experiments for classical super-resolution, deblurring, colorization, non-align JPEG compression, and bad weather restoration show the effectiveness of the proposed method.

**Strengths:**

- The paper works on an important problem: unsupervised image restoration.
- The paper proposes a method to drive the latent variables to a high probability set to generate high-quality restoration results and provide theatrical analysis.
- The proposed method generalizes to non-linear degradation and bad weather as well as applies to diffusion models with latent space, e.g., Stable Diffusion.
- Well written and easy to follow.

**Weaknesses:**

- For the proposed variance preservation sampling (VPS), the latent variables fall into the high probability set with a strong assumption that $M$ is sufficiently large (Theorem 2.2). However, $M$ is empirically set to $1$ in this paper, and there is no detailed discussion about this important parameter.
- The qualitative results are only shown for the proposed method, and there are no visual comparisons to existing methods.
- For the bad weather degradation, there are no quantitative measurements. The only evidence is a set of visual examples in Fig. 2.
- The ablation study is only conducted on one type of degradation and is insufficient.
- The paper should discuss and compare with more recent blind image restoration works, e.g., GDP.

Fei, Ben, et al. "Generative Diffusion Prior for Unified Image Restoration and Enhancement." CVPR. 2023.

**Questions:**

- Is the proposed method sensitive to some parameters, e.g., $\gamma$ and strength?
- The paper should give more visual illustrations for the sampling algorithm of different timesteps to understand the effectiveness of the two steps in the proposed VPS.
- For the bad weather, the training of diffusion models often encounters such images in high image quality. It is not evident that the images with bad weather are categorized as degenerated or clean from the perspective of the diffusion prior.
- The authors should provide more implementation details.
- It is better to explain $x$ and $y$ in Fig. 2.

---

> ### Author Response · Authors · 2023-11-19
> **Response to Reviewer HBur**
>
> **Q:** For the proposed VPS, the latent variables fall into the high probability set when M is sufficiently large (Theorem 2.2). However, M is empirically set to 1 in this paper, and there is no detailed discussion about this important parameter.
>
> **A:**  $M=1$ works well because VPS is conducted for each timestep. This does not conflicts with Theorem 2.2 which states that when $M$ is sufficiently large, latents converges to a High Probability Set for certain timestep $t$. In other words, **the correcting effect of VPS can be accumulated along the DDIM sampling trajectory**. We have conducted noisy SR experiments on CelebA with different $M$ and the tendency of performance is plotted in Figure A2. The results suggests that increasing $M$ cannot significantly boost performance but with more compute. Therefore, we set $M=1$ for most cases.
>
> **Q:** There are no visual comparisons to existing methods.
>
> **A:** We follow the reviewer's advice to add visual comparisons with existing methods. Figures A7 and A8 presents visual comparisons of noisy SR and JPEG artifacts correction respectively. Figure A9 shows comparisons of $32$ SR with $\sigma=0.1$ using Stable Diffusion XL. It can be found that without knowing degradation model, DreamClean (ours in Figures A7, A8 and A9) can improve images' quality and produce visually pleasing results. When equipped with specific degradation, DreamClean (ours* in Figure A7) yields more faithful results.
>
> **Q:** For the bad weather degradation, there are no quantitative measurements. The only evidence is a set of visual examples in Fig. 2.
>
> **A:** We follow the reviewer's advise and include quantitative scores of image deraining on Rain100L. Table 1 shows the quantitative results. Baseline refers to original unprocess data. It can be found that without any specific training, DreamClean can effectively process complex rainy degradation.  We also add more bad weather corrupted cases. Figures 2 and A12 present more bad weather degradation cases. Besides, Figure A1 presents more real-world bad weather cases.
>
> **Table 1: Quantitative results on Rain100L.**
> |Rain100L|PSNR$\uparrow$|SSIM$\uparrow$|LPIPS$\downarrow$|
> |:----|:----:|:----:|:----:|
> |Baseline|28.53|0.89|0.06|
> |Ours|28.68|0.92|0.04|
>
> **Q:** The ablation study is only conducted on one type of degradation.
>
> **A:**
> We follow the suggestion to perform ablation experiments on more degradation types. The below table shows that our VPS achieves the best score compared with other schemes. The reason is that VPS optimizes latents to High Probability Set, which conforms with the sampling dynamics of diffusion models. These results are consistent with noisy SR cases.
>
> **Table 2: Ablation study on more degradation types.**
> |Schedule|Deblurring(Gauss) PSNR$\uparrow$/SSIM$\uparrow$/LPIPS$\downarrow$|Inpainting PSNR$\uparrow$/SSIM$\uparrow$/LPIPS$\downarrow$|
> |:----|:----:|:----:|
> |0|32.20/0.85/0.12|24.24/0.83/0.10|
> |$\sqrt{2\gamma(1-\bar{\alpha}_t)}$|32.68/0.90/0.10|24.67/0.85/0.09|
> |Ours|32.71/0.91/0.09|24.71/0.87/0.07|
>
> **Q:** The paper should discuss and compare with more recent blind image restoration works, e.g., GDP.
>
> **A:** We follow the reviewer's suggestion to add comparison with GDP. For reviewer's convinience, we copy the quantitative results of noisy $4\times$ super-resolution on imagenet and CeleBA.
>
> **Table 3: Quantitative comparison with GDP.**
> |CeleBA|PSNR$\uparrow$|SSIM$\uparrow$|LPIPS$\downarrow$|NFEs$\downarrow$|
> |:----|:----:|:----:|:----:|:----:|
> |GDP|24.38|0.71|0.15| 1000|
> |Ours| 27.23|0.77 |0.12|90|
> |Ours*| 30.19| 0.84 | 0.08|60|
>
> |Imagenet|PSNR$\uparrow$|SSIM$\uparrow$|LPIPS$\downarrow$|NFEs$\downarrow$|
> |:----|:----:|:----:|:----:|:----:|
> |GDP|24.33|0.67|0.39|1000|
> |Ours|24.31|0.67|0.40|90|
> |Ours*|25.84|0.74|0.23|60|
>
> **Q:** Is the proposed method sensitive to some parameters, e.g., step size $\gamma$ and strength $\tau$?
>
> **A:** The effective range of $\gamma$ and $\tau$ is relatively loose. Empirically, we find that $\gamma \in [0.01, 0.1]$ and strength $\tau \in [300, 500]$ works well in most cases. To validate this, we conduct noisy SR experiments on CelebA with a range of $\gamma$ and strength $\tau$. As shown in Figures A3 and A4, our method works well when $\gamma$ and $\tau$ are in the above interval.

---

> > ### Author Response · Authors · 2023-11-19
> > **More Response**
> >
> > **Q:** The paper should give more visual illustrations for the sampling algorithm of different timesteps to understand the effectiveness of the two steps in the proposed VPS.
> >
> > **A:** We follow the reviewer's advice and add visual illustration in Figure A5. The visualization in Figure A5 intuitively illustrates the respective function of the VPS and DDIM step respectively. We can find that after VPS correction, the original degraded artifatc is translated to Gaussian-like noise. Therefore, VPS step is responsiable for correcting the corrupted low-probability latents. Moreover, after DDIM step, the amount of noise is decreased progressively. Thus, DDIM step is responsible for progressively reducing the amount of Gaussian noise contained in latents.
> >
> > **Q:** For the bad weather, the training of diffusion models often encounters such images in high image quality. It is not evident that the images with bad weather are categorized as degenerated or clean from the perspective of the diffusion prior.
> >
> > **A:** It is a very practical concern. We agree that training data containing bad weather corrupted images may make the diffusion hard to distinguish degraded or clean cases. Ideally, DreamClean should require a diffusion prior of clean images, which in turn requires a diffusion model should pretrained on high-quality clean images. Therefore, if DreamClean is used to tackle with bad weather cases, it is desirable to exclude such images from training data of diffusion models.
> > In this work, DreamClean use the pre-trained diffusion models to distinguish clean and bad weather cases because the training data (e.g., ImageNet and CelebA) of diffusion models seldom contains bad weather corrupted images. That is, the probability of sampling clean images is much higher than sampling bad weather corrupted images from the pretrained diffusion models. Therefore, we can anticipate DreamClean can remove bad weather artifacts by increasing the probability.
> >
> >
> > **Q:** The authors should provide more implementation details.
> >
> > **A:** Thanks for the reminder. We additionally provide the intact DreamClean algorithm in Algotihm 3 (Page 19) by integrating original VPS of Algorithm 2 and DDIM inversion in Algorithm 1. We also add detailed comments to all components. We set $\tau=300, M=1,$ and $\gamma=0.05$. The corresponding $\eta_g$ and $\eta_l$ can be computed by $\eta_l=\gamma(1-\bar{\alpha}_t)$ and $\eta_g=\sqrt{2\gamma(1-\bar{\alpha}_t)}$. $\bar{\alpha}_t$ is the noise schedule determined by pretrained diffusion models.
> >
> > **Q:** It is better to explain x and y in Fig. 2.
> >
> > **A:** $\textbf{y}$ means the degraded image and $\textbf{x}$ means the processed result of DreamClean. We add proper explainations to avoid confusion.

---

> > > ### Author Response · Authors · 2023-11-22
> > > **Eager to hearing from you!**
> > >
> > > Dear Reviewer HBur,
> > >
> > > As you may notice the ICLR rebuttal discussion period is near the end. We would like to know if you still have any concerns regarding our work and the response. If you still have any concerns, we are eager to hear from you, please let us know and we are more than happy to discuss them with you. If all your concerns have been well addressed, we sincerely wish you to raise the rating to reflect that. Your support is really important for our work.
> > >
> > > Best wishes,
> > >
> > > ICLR Submission 331 Authors.

---

> > > > ### Comment · Reviewer_HBur · 2023-11-22
> > > > **Official Comment by Reviewer HBur**
> > > >
> > > > I sincerely thank the authors for the elaborate responses to my questions. Nevertheless, I still have some concerns.
> > > > For the statement "the correcting effect of VPS can be accumulated along the DDIM sampling trajectory," is there any theoretical analysis?
> > > > It is better to include GDP for visual comparison.
> > > > For Figure A1 (a), the hallucination, deviation from the original contents, in bad weather is more evident than in other applications. Is there any trade-off between the restoration and regeneration?

---

> ### Author Response · Authors · 2023-11-23
> **Response to Reviewer HBur**
>
> Thanks for your feedback! Below are our detailed responses.
>
> **Q:** The correcting effect of VPS.
>
> **A:** The form of VPS: $y_t^m = y_t^{m-1}+\eta_l\nabla\log p_t(y_t^{m-1})+\eta_g\epsilon_g^m$. We consider the degraded input consists of clean and corrupted components.  We assume that it is the corrupted component that causes the low probability of latents and the amount of corrupted information remains constant.
> For arbitrary $t$, the first term $\eta_l\nabla\log p_t(y_t^{m-1})$ aims to increase its probability by removing low-probability corrupted component and the second term $\eta_g\epsilon_g^m$ adds a finely tuned amount of Gaussian noise to ensure that $y_t$ adheres to the formulation of diffusion models.
> It is worth noting that the addition of fresh Gaussian noise does not contribute to an increase in the level of corrupted information. This is because the amount of noise is carefully adjusted by $\eta_g$ to ensure that it can be totally removed after the sampling process of diffusion models.
> Therefore, in essence, VPS exchanges low-probability corrupted component for Gaussian noise, The amount of exchange is adjusted to match the dynamics of diffusion models. As this exchange occurs along the ODE trajectory and the total amount of corrupted information remains constant, the correcting effect accumulates gradually.
> To support this, please refer to visualization in Figure A5. Comparing latents of the interval $t\in [240, 400]$ of Figure a (without VPS) and Figure b (with $M=1$ VPS), we can observe the accumalated correcting effect. That is, at $t=400$, artifacts are still apparent after VPS. After accumulation along the sampling trajectory, at $t=240$, the artifacts are significantly reduced and the latent is like the clean image perturbed with Gaussian noise, which conforms the formulation of diffusion models. Additionally, we provide a theoretical analysis in Sec. A. 12 (Page 21).
>
>
> **Q:** Include GDP for visual comparison.
>
> **A:** We follow the advice to include GDP for visual comparison.
>
>
> **Q:** The hallucination in bad weather.
>
> **A:** The more hallucination in real-world bad weather cases can be explained that the degradation model is more complex and unknown. Therefore, the coupling of clean and corrupted component of the input image becomes more intricate in such scenarios. As we discuss in Limitation section (Secion 5, Page 9), DreamClean cannot guarantee strict faithfulness, i.e., without the knowledge of the degradatin model, DreamClean cannot provide complete separation between the clean and corrupted components of the input image.
> DreamClean provides a ''soft'' separation from the probabilistic prespective: the corrupted component should be evaluated as low probability using the metric of the generative prior learned from clean images. Therefore, the hallucination emerges because DreamClean resamples the low-probability component according to its generative prior learned from clean training images. The mentioned trade-off exists and can be implemented by the step size $\gamma$. In the limit that $\gamma=0$, VPS vanishes and DreamClean reproduces the input image using the DDIM inversion. As $\gamma$ increases, DreamClean regenerates the low-probability component.

---

### Official Review · Reviewer_NQCS · 2023-10-30

**Soundness:** 4 excellent
**Presentation:** 3 good
**Contribution:** 4 excellent
**Rating:** 8
**Confidence:** 5

**Summary:**

The paper proposes a novel unsupervised diffusion based approach called DreamClean, which does not rely on specific degradation types, for Image Restoration tasks. The superiority lies in restoring clean images without specific finetuning or assuming known degradation. Specifically, DreamClean exploits the approximately inversible property of DDIM to ensure faithfulness. For realness, the paper introduces Variance Preservation Sampling (VPS) to guide latents to nearby high probability region which conforms statistics of the pretrained diffusion model to produce high-quality images. The paper gives a solid theoretical support for the convergence of VPS. The experiments and analysis are sufficient and the results seems promising.

**Strengths:**

- The paper gives a simple but insightful conclusion for the existing IR methods and points out the advantage of unsupervised method for generalization. Based on this, DreamClean is proposed to mitigate the reliance on the underlying degradation model or data-specific finetuning, which is critical to improve the generalization and the practical.
- DreamClean can improve image quality using pre-trained diffusion models without finetuning or assuming specific degradation formulation. This method fills this gap in IR field. The results on various degradation types and latent diffusion model seems to support its strong robustness.
- The analysis of the paper is mathematically complete. The paper constructs a high probability set according to the diffusion’s statistical characteristic and proves that latents will converge to the high probability set under VPS. This perspective is interesting and novel.
- The experiments are sufficient enough to support the claims and the results are promising. In particular, the result in Figure 1 has much higher visual quality compared with previous methods. The paper validates the efficacy on various IR tasks including single and multiple degradation cases, and well compatibility with latent diffusion models.

**Weaknesses:**

- The proposed method leverages ODE inversion to ensure faithfulness with motivation that ODE can approximately reconstruct the input image. Given that, why is ODE inversion needed rather than directly conduct VPS on the input image y?
- The authors should explain the reason that fresh noise is need in VPS. In other words, without the noise, VPS returns latents with locally maximal probability density. Why do higher-density latents perform inferior compared to latents with noise?

**Questions:**

- In section 3.2, the authors should give a more detailed interpretation about  log⁡(q(x_t |x_0)) can be alternative score for log⁡(p_θ (x_t)).

---

> ### Author Response · Authors · 2023-11-19
> **Response to Reviewer NQCS**
>
> **Q:** Why is ODE inversion needed rather than directly conduct VPS on the input image y?
>
> **A:** ODE inversion indeed perturbs the data with noise. As revealed in Song et at [1], perturbing data can effectively mitigate the issue of inaccurate score estimation in low data density regions. With progressively decreasing noise, high-quality images can be generated. To further support this,  experiments of noisy SR based on CelebA are conducted. Table 1 reveals that inversion provides more accurate score estimation, thereby boosting the final IR performance.
>
> **Table 1: Performance tendency with different ODE inversion strength.**
> |$\tau$|10|100|300|500|
> |:----|:----:|:----:|:----:|:----:|
> |PSNR$\uparrow$|23.65|24.24|27.23|27.25|
> |SSIM$\uparrow$|0.51|0.57|0.77|0.77|
>
> **Q:** The authors should explain the reason that fresh noise is need in VPS. Why do higher-density latents perform inferior compared to latents with noise?
>
> **A:** For certain timestep $t$, according to diffusion's formulation, latents should locate in the High Probability Set. As we state in Theorem 2.2, the amount of fresh noise is vital to ensure that the corrected latents is driven to the High Probability Set. Therefore, fresh noise is needed to conform the sampling dynamics of diffusion models, which produces high-quality images. Besides, as shown in Figure 8, without fresh noise ($\eta_g=0$), many details are discarded in the generated image.
>
> [1] Generative modeling by estimating gradients of the data distribution. NeurIPS 2019.

---

> > ### Author Response · Authors · 2023-11-19
> > **More Response**
> >
> > **Q:** $\log⁡(q(\mathbf{x_t} |\mathbf{x_0}))$ can be alternative score for $\log(p_\theta (\mathbf{x_t}))$.
> >
> > **A:** Since diffusin process is reversiable, we have $\log p_\theta(\mathbf{x_t})\approx \log q(\mathbf{x_t})=\int \log q(\mathbf{x_t}|\mathbf{x_0})q(\mathbf{x_0})\rm d \mathbf{x_0}$. Therefore, when $\mathbf{x_0}$ is given, we can choose $\log q(\mathbf{x_t}|\mathbf{x_0})$ as the alternative score for $\log(p_\theta (\mathbf{x_t}))$.

---

### Official Review · Reviewer_aVXX · 2023-10-31

**Soundness:** 3 good
**Presentation:** 3 good
**Contribution:** 3 good
**Rating:** 6
**Confidence:** 5

**Summary:**

This paper presents DreamClean, a training-free method for high-fidelity image restoration without prior knowledge of degradation. DreamClean embeds the degraded image into pre-trained diffusion models, uses Variance Preservation Sampling (VPS), and outperforms previous methods, especially in challenging tasks without degradation priors.

**Strengths:**

1. The innovation is impressive, providing a fresh perspective on the field of image restoration。

**Weaknesses:**

1. Insufficient experiments. The performance needs to be tested on a broader range of real-world degradation data. The data used in the paper are all synthetic, which lacks convincing power.
2. The methods for comparison are insufficient. For example, other training-free methods like 'Generative Diffusion Prior for Unified Image Restoration and Enhancement' are also available.

**Questions:**

Refer to weaknesses.

---

> ### Author Response · Authors · 2023-11-19
> **Response to Reviewer aVXX**
>
> **Q:** The performance needs to be tested on a broader range of real-world degradation data.
>
> **A:** We follow the reviewer's suggestion to add more experiments on real-world degraded images. First, we evaluate quantitative performance of image denoising on the real-world SIDD dataset [2]. For comparison, we include two baselines: the scores of input noisy image and the classic denoising method BM3D [1]. Table 1 reveals the efficacy of DreamClean in real-world image denoising ($+8.26$ dB compared with Input).
> Besides, we also extend DreamClean to more real-world applications, ranging from real-world image deraining, desnowing and old photo restoration and real-world image denoising. As show in Figure A1, DreamClean can process complex real-world corrupted images. We have added related content in Sec. A.3.
>
> **Table 1: Quantitative results of real-world image denoising on SIDD.**
> |Method|PSNR$\uparrow$|SSIM$\uparrow$|LPIPS$\downarrow$|
> |:----|:----:|:----:|:----:|
> |Input|23.66|0.35|0.58|
> |BM3D|25.65|0.68|N/A|
> |Ours|31.92|0.76|0.23|
>
> **Q:** DreamClean should compare with recent training-free method GDP.
>
> **A:**: We follow the reviewer's advice and add comparison with GDP. For reviewer's convinience, we copy the quantitative results of noisy SR on Imagenet and CeleBA.
>
> **Table 2: Quantitative comparison with GDP.**
> |CeleBA|PSNR$\uparrow$|SSIM$\uparrow$|LPIPS$\downarrow$|NFEs$\downarrow$|
> |:----|:----:|:----:|:----:|:----:|
> |GDP|24.38|0.71|0.15|1000|
> |Ours|27.23|0.77|0.12|90|
> |Ours*|30.19|0.84|0.08|60|
>
> |Imagenet|PSNR$\uparrow$|SSIM$\uparrow$|LPIPS$\downarrow$|NFEs$\downarrow$|
> |:----|:----:|:----:|:----:|:----:|
> |GDP|24.33|0.67|0.39|1000|
> |Ours|24.31|0.67|0.40|90|
> |Ours*|25.84| 0.74|0.23|60|
>
> [1] Image Denoising by Sparse 3-D Transform-Domain Collaborative Filtering. TIP 2007.
>
> [2] A High-Quality Denoising Dataset for Smartphone Cameras. CVPR 2018.

---

> ### Author Response · Authors · 2023-11-22
> **Eager to hearing from you!**
>
> Dear Reviewer aVXX,
>
> The ICLR rebuttal discussion period is near the end. Does our response successfully address all your concerns? If so, would you please raise your rating? Your support is really important to us. If not, do please tell us your remained concerns. We are eager to discuss them with you.
>
>
> Best wishes,
>
> ICLR Submission 331 Authors.

---

### Author Response · Authors · 2023-11-19
**Global Response**

Thanks for the reviewer's time and efforts. We appreciate the valuable comments. We have carefully reviewed the suggestions and have made the necessary revisions to improve the paper. The newly added content is highlighted in blue color for better visibility, which is summarized as:
* Add extension to real-world applications in Sec A.3, including quantitative results on SIDD in Table A1 and visualizations in Figure A1;
* Add comparison with GDP in Tables 1 and 2 and visual comparison with other method in Figures 7,8 and 9.
* Add analysis about hyperparameters in Sec. A4, including Figures A2-A4;
* Add visualization of DDIM and VPS in Sec. A5;
* Add results on Rain100L in Sec. A6 and more ablation study in Sec. A9;
* Add the intact algorithm in Sec. A7.

---

### Meta-Review · Area_Chair_UCs2 · 2023-12-11

**Metareview:**

All the reviewers are positive about the paper: DreamClean can improve image quality without relying on specific degradation models or data-specific finetuning; the model is compatible with latent diffusion models and hence harness  all the advantages; the analysis of the paper is thorough with an interesting and novel perspective on latents converging to a high probability set under variance preservation sampling (VPS), improving the likelihood of yielding clean images; the experiments are well-conducted, supporting the claims with impressive results.

The reviewers also shared weaknesses: the performance of DreamClean is primarily tested on synthetic data, which lacks convincing power for real-world applicability; there is a lack of comparison with other contemporary models on common degradation scenarios; the necessity of ODE inversion and the inclusion of fresh noise in VPS lack clear explanation. Also, dependence on pre-trained models can lead to generation of unexpected images due to dataset biases.

During the rebuttal, the authors provided detailed additional experiments to address many of the concerns from the reviewers. As a result, the paper received two 6 and two 8 ratings. The AC also feels that the contribution of the paper is solid and warrants publication.

**Justification For Why Not Higher Score:**

While I don’t mind the paper being bumped up to the spotlight, the heavy lifting of this work is done by high-quality, pretrained stable diffusion models, and is also biased by the behavior of these diffusion models.

**Justification For Why Not Lower Score:**

All the reviewers are positive about the paper. The contribution is solid and the results are impressive.

---

### Decision · Program_Chairs · 2024-01-16

Accept (poster)